# Comment on "Transport of substantial stratospheric ozone to the surface by a dying typhoon and shallow convection" by Chen et al. (2022)

Xiangdong Zheng[1], Wen Yang[2], Yuting Sun[1,3],Chunmei Geng[2], Yingying Liu[2], ,Xiaobin Xu[1]

[1]Chinese Academy of Meteorological Sciences, Beijing 100081, China

[2]State Key Laboratory of Environmental Criteria and Risk Assessment, Chinese Research Academy of Environmental Sciences, Beijing 100012, China
[3]Nanjing University of Information Science & Technology, Nanjing, Jiangsu, 210044, China

*Correspondence to*: Xiaobin Xu (xiaobin_xu@189.cn)

**Abstract.** Chen et al. (2022) analyzed the event of rapid nocturnal $O_3$ enhancement (NOE) observed on 31 July 2021 at surface level in the North China Plain and proposed transport of substantial stratosphere ozone to the surface by Typhoon In-fa followed by downdraft of shallow convection as the mechanism of the NOE event. The analysis seems to be valid in the view-point of atmospheric physics. This comment revisits the NOE phenomenon on the basis of the China National Environmental Monitoring Center (CNEMC) network data ever used in Chen et al. (2022), together with the CNEMC data from Zibo (ZB), and $O_3$, $NO_x$, PAN (peroxyacetic nitric anhydride) and VOCs (volatile organic compounds) data from the Zibo supersite operated by China Research Academy of Environmental Sciences (CRAES). We found (a) $O_x$ ($O_3+NO_2$) levels during the NOE period approaching to those of $O_3$ during 14:00-17:00 LT; (b) the levels of PAN and the relationship between $O_3$ and PAN consistent with dominance of chemical and physical processes within the boundary layer, and (c) estimated photochemical ages of air mass being shorter than one day and showing no drastic increases during the NOE. We argue that the NOE was not caused by typhoon-induced stratospheric intrusion but originated from fresh photochemical production in the lower troposphere. Our argument is well supported by the analysis of atmospheric transport as well as ground-based remote sensing data.

## 1 Introduction

Chen et al. (2022) reported a phenomenon of rapid nocturnal ozone ($O_3$) enhancement (NOE) occurred at the surface level during the night of 31 July 2021 in six cities in the North China Plain (NCP, 34-40 °N, 114-121 °E). Prior to the NOE, the NCP was impacted by Typhoon In-fa, which was largely weakened by 30 July 2021. The mesoscale convective systems (MCSs) formed and passed through the NCP at night on 31 July 2021. Chen et al. (2022) concluded that the NOE phenomenon resulted from "the direct stratospheric intrusion to reach the surface" and was "induced by the multi-scale interactions between the dying Typhoon In-fa and local MCSs". The study suggested that the dying Typhoon In-fa induced stratospheric troposphere transport (STT) of $O_3$ followed by downdrafts of shallow convections, which resulted in "transport

of substantial stratospheric ozone to the surface". The relatively high $O_3$-low water vapor and CO (HOLWCO) concentrations observed at some sites in the NCP and the relative variations of water vapor, $O_3$ profiles respectively from radiosonde data and the AIRS satellite product were used to support the conclusions.

STT processes are triggered by the large-scale circulation or synoptic-scale dynamical processes (Holton et al., 1995). A global study (Škerlak et al., 2014) shows that STT displays strong regional distribution and seasonal variations, and the NCP is not a hot region particularly in summer. STT can be an important source of tropospheric $O_3$, particularly in regions where the photochemistry is weak (Lelieveld and Dentner, 2000). However, tropospheric $O_3$ originates dominantly from photochemistry within the troposphere and photochemically produced $O_3$ (PPO) is the more important $O_3$ source not only in the middle-low troposphere (Lelieveld and Dentener, 2000; Logan, 1985) but also in the upper troposphere (Chameides, 1978; Liu et al., 1983; Jaeglé et al., 1998). Anthropogenic and natural $O_3$ precursors convectively transported from the surface or lower troposphere and lightning produced $NO_x$ may involve PPO in the upper troposphere. Precursors emitted near the surface contribute largely to PPO in the surface and boundary layer, which can be transported upwards through the warm conveyor belt (Bethan et al., 1998; Cooper et al., 2002), spread in the free atmosphere and delivered over a long range by atmospheric circulations (Parrish et al., 1998) . PPO in the surface boundary layer is mainly removed by $NO_x$ titration reactions and dry deposition. The NCP is a hot region of PPO from the surface level up to 2.5 km in the summer (e.g., Ding et al., 2008) and has demonstrated rapid long-term increases in surface $O_3$ levels (Ma et al., 2016; Lu et al., 2018; Lyu et al., 2023).

Typhoons are tropical cyclones formed over the western North Pacific regions, which have well-organized structures of updrafts and downdrafts over hundreds and thousands of kilometres (Ahrens and Henson, 2016). A large–scale tropical cyclone with well self-organized character is able to induce dynamical processes and form an outflow layer in the upper troposphere and lower stratosphere (UTLS) and cause strong downdrafts in the periphery of the cyclone (Merrill, 1988; Ahrens and Henson, 2016). The strong air subsidence in the periphery of a typhoon can theoretically lead to the STT of $O_3$. It was suggested that the observed enhancement of $O_3$ in the middle troposphere over the Indian Ocean was caused by the STT through ageostrophic process linked to the strong tropical cyclone Marlene, which occurred in April 1995 (Baray et al., 1999). This suggestion is supported by a modeling study (de Bellevue et al., 2007). However, the idea that $O_3$ increases in the mid- and upper-troposphere are directly from the STT processes induced by typhoons is less supported by in-situ aircraft-borne observations by Cario et al. (2008) and the literature reviewed therein. Especially, the comparative studies on the Supertyphoon Mireille (1991) during the Pacific Exploratory Mission(PEM)-West A campaign (Newell et al., 1996a; Preston et al., 2019) and the Hurricanes Floyd (1999) and Georges (1998) in the Atlantic Ocean during their phases of intensification and weakening (Carsey and Willoughy, 2005) provided little evidence of the STT of $O_3$. On the other hand, the analysis of ERA5 PV (potential vorticity) and air mass with HOLWCO observed below 12 km by the In-service Aircraft for a Global Observing System (IGOS) indicated the occurrence of STT induced by typhoons (Roux et al., 2020; Z. Chen et al., 2021). However, it should be noted that atmospheric large-scale subsidence over East Asia can also be induced by the

strong summer subtropical high. Photochemically aged pollution air masses may also show HOLWCO features as observed in the PEM-West A campaign (Newell et al. 1996b; Stoller et al., 1999).

## 2 Data

Surface $O_3$, $NO_x$, and meteorology data collected in the cities Hengshui (HS), Binzhou (BZ), Jinan (JN), Weifang (WF), Qingdao (QD), Weihai (WH) and the newly added Zibo (ZB) were from the China National Environmental Monitoring Center (CNEMC) network (https://quotsoft.net/air). The geographical location of ZB is shown in Figure S1. In addition, hourly averages of surface $O_3$, PAN (peroxyacetic nitric anhydride), $NO_x$, and VOCs (volatile organic compounds) were obtained from a supersite in ZB, operated by China Research Academy of Environmental Sciences (CRAES). Ambient $O_3$ and $NO_x$ at the supersite were monitored using a Model 49i ozone analyzer and a Model 42i $NO/NO_2/NO_x$ analyzer (both from Thermo Fischer Scientific), respectively. The analyzers were calibrated weekly. Quasi-continuous measurement of PAN was made using a gas chromatograph coupled with an electron capture detector (GC-ECD) (ZC-PANs, Research Center for Eco-Environmental Sciences, Chinese Academy of Sciences). The GC-ECD system was calibrated seasonally using PAN inline produced from $CH_3COCH_3$+NO reactions under UV irradiation (J. Chen et al., 2021). Samples of VOCs were taken hourly and analyzed using a coupled gas chromatograph-mass spectrometry (GC-MS) system (5800-GM, Thermo Fischer Scientific), which was calibrated monthly using standard gas mixture from Linda, containing 116 species including hydrocarbons, oxygenated and halogenated hydrocarbons.

## 3 Verification of results in Chen et al. (2022)

### 3.1 Summary of observed NOE events

The hourly averages of surface $O_3$ and $NO_x$ in the six cities (HS, BZ, JN, WF, QD, and WH) listed in Chen et al., (2022) as well as those from ZB and the ZB supersite are shown in Figure 1. The data of $O_3$ and $NO_x$ from 18:00 LT on 31 July to 06:00 LT on 1 August 2021 are highlighted in red lines in Figures 1 and 2S. Although PPO was not obvious during 29-30 July due to the weather conditions, the diurnal variations of $O_3$ and $NO_x$ on most days displayed typical features being controlled by the PPO process, with $O_3$ maxima and correspondingly $NO_x$ minima during 14:00~17:00 and rapid nighttime $O_3$ decreases due to substantial $NO_x$ titration reactions and dry deposition. As reported by Chen et al. (2022), a clear NOE was observed during the night of 31 July in HS, BZ, JN, WF, and QD. Our data from ZB (Figure 1g and h) also confirm the occurrence of this NOE. However, it is noteworthy that NOE events occurred not only during the night of 31 July but also during some other nights in these cities with $O_3$ enhancement of 5-20 ppbv. The frequency of NOE was highest in WH, in details: $O_3$ increased from 24±8 ppbv at 22:00 LT on 27 July to 46±21 ppbv at 01:00 LT on 28 July; from 23±5 ppbv at 23:00 LT on 29 July to 40±4 ppbv at 04:00 LT on 30 July; from 43±2 ppbv at 00:00 LT to 57±3 ppbv at 04:00 LT on 31 July; from 54±19 ppbv at 00:00 LT to 66±5 ppbv at 02:00 LT on 2 August; from 42±17 ppbv at 00:00 LT to 48±5 ppbv at 03:00

LT on 3 August; from 20±4 ppbv at 02:00 LT to 25±2 ppbv at 03:00 LT on 5 August. Other NOE events occurred, for
example, on 4 August in HS, on 28 July and 5 August in BZ, on 26 July and 4 August in JN, on 4 August in WF and QD.
Therefore, the NOE events occur frequently in the NCP, regardless the impacts from typhoon or STT, as already reported in
He et al. (2022).

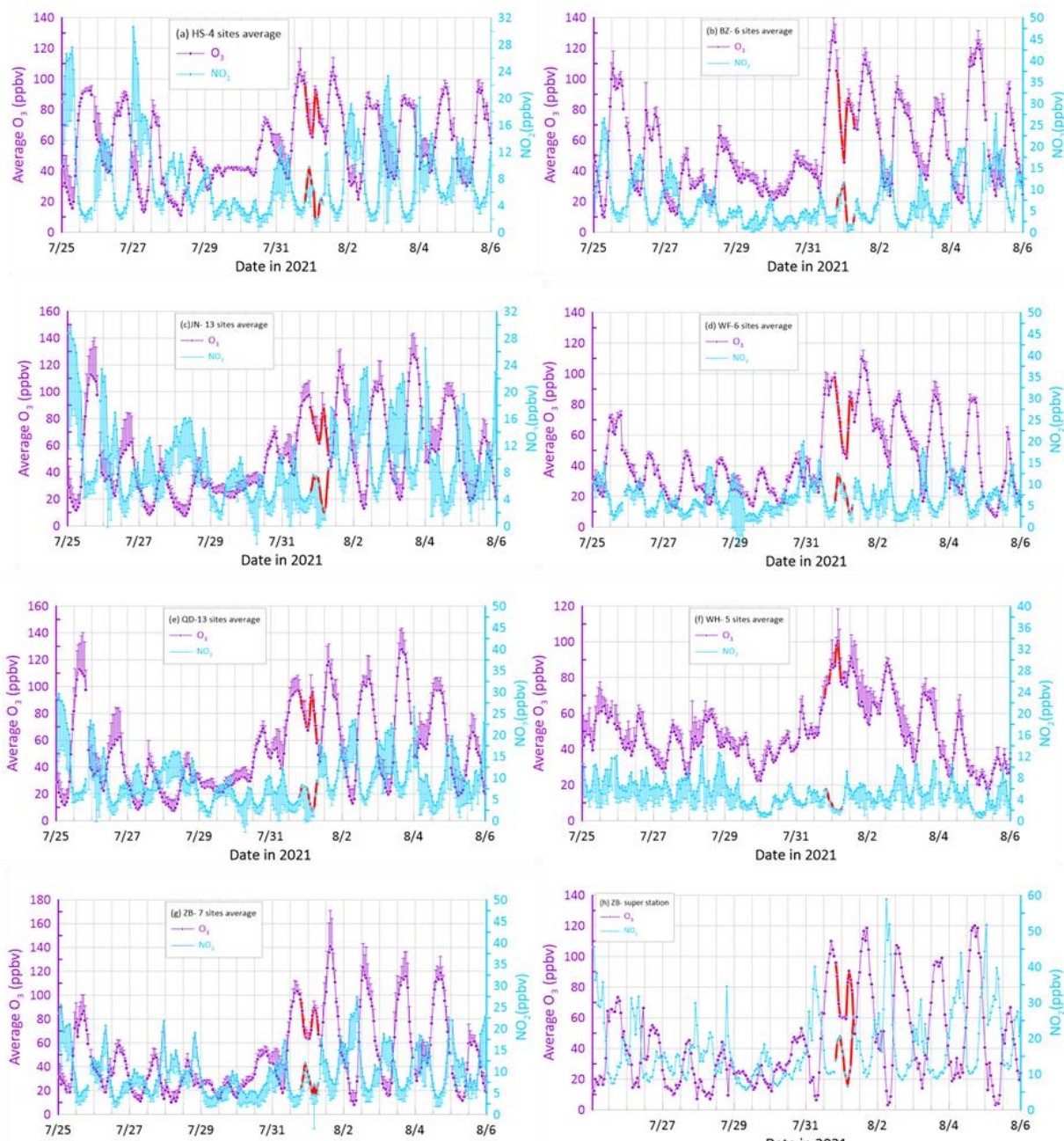

Figure 1: Time series of hourly multi sites-average of O$_3$ (purple) and NO$_2$ (bright blue) in several NCP cities between 25 July and 5 August 2021. Data from 18:00 LT on 31 July to 06:00 LT on 1 August are highlighted in red. The multisite data from HS (4 sites) (a), BZ (6 sites) (b), JN (13 sites) (c), WF (6 sites) (d), QD (13 sites) (e), WH (5 sites) (f) and ZB (7 sites) (g) are available at https://quotsoft.net/air (last access: 15 April 2023; X. L. Wang, 2020). Data from the ZB supersite (h) is provided by the Chinese Academy of Environmental Sciences (CRAES). The positive (negative) error bars represent one standard deviation of O$_3$ (NO$_2$).

## 3.2 Identifying the origin of the NOE by comparing afternoon $O_3$ with $O_x$ in the NBL

Following the method of He et al. (2022), we make comparison of $O_3$ averages during 14:00-17:00 LT on 31 July in the above cities with the respective $O_x$ ($O_3+NO_2$) averages during the periods of the maximum NOE between 31 July and 1 August (Table 1). Such comparison facilitates the judgment whether or not the NOE was caused by downward mixing of air in the residual layer (RL) into the nocturnal boundary layer (NBL) because afternoon averages of $O_3$ in the convective boundary layer are well preserved at night in the RL and $O_x$ is a more conserved quantity than $O_3$ in the NBL. Details about the reasonability of this method are given in Text S1. It can be seen in Table 1 that, except for WH, the nighttime $O_x$ averages approach to or obviously lower than the respective daytime $O_3$ averages. In the mega-cities QD and JN, the average levels of $O_x$ during the maximum NOE were nearly the same as those of daytime $O_3$, while the nighttime $O_x$ in the other cities (excluding WH) was at least a few ppbv lower than daytime $O_3$. According to the discussions in Text S1, for all cities excluding WH, data in Table 1 do not suggest any significant STT impact on the NOE events.

Table 1: Averages of surface $O_3$ during 14:00-17:00 LT and $O_x$ during the periods of the maximum NOE in the night from 31 July to 1 August 2021.

| Cities | NOE time | Daytime mean $O_3$±std (ppbv) | Nighttime mean $O_x$±std (ppbv) |
|---|---|---|---|
| HS (4 sites) | 01:00-02:00 | 101.7±3.5 | 93.1±2.2 |
| BZ(6 sites) | 01:00-03:00 | 124.1±5.4 | 94.6±2.3 |
| JN(13 sites) | 02:00-03:00 | 96.1±1.6 | 96.6±2.2 |
| WF(6 sites) | 04:00-05:00 | 92.5±4.8 | 88±0.3 |
| QD(13 sites) | 02:00-03:00 | 96.1±1.6 | 96.6±2.2 |
| WH(5 sites) | 02:00-03:00 | 59.2±7.1 | 101.3±1.9 |
| ZB(7 sites) | 02:00-05:00 | 102.6±1.4 | 90.9±4.2 |
| ZB-Supersite | 02:00-05:00 | 104.2±4.3 | 93.8±3.0 |

## 3.3 Analysis of atmospheric transport

The case of WH deserves more detailed analysis. WH is a relatively smaller city in the tip of the Shandong Peninsula. The higher $O_x$ concentration for WH in Table 1 was probably related to the regional transport of air pollution and the influence of the diurnal alternations of land-sea breezes. Actually, the afternoon PPO in WH had not been well established as shown in Figure 1f. During the daytime (particularly afternoon), when sea breeze dominates, PPO is significantly diluted by cleaner air from the marine boundary layer. The daily maximum of $O_3$ in WH is generally observed between 11:00 LT and 13:00 LT, rather than between 14:00 LT and 17:00 LT. If the $O_3$ average for WH in Table 1 were replaced with that from 11:00 LT to

13:00 LT on 1 August (80 $\pm$ 2.0 ppbv), then the difference between $O_x$ and $O_3$ would be reduced to about 21 ppbv. At night, when the land breeze dominates, the near surface level of WH is usually controlled by divergence, which induces downdraft from the residual layer, transports daytime PPO residing in the residual layer to the surface, and resulted in the NOE. This might have been the main reason of highly frequent NOE emerging in WH.

Chen et al. (2022) investigated the atmospheric transport process of the NOE by using high-resolution Weather Research and Forecasting (WRF) simulation and FLEXible PARTicle (FLEXPART) particle dispersion modelling. They presented the two scenarios for BZ and QD and found very different results for the two cities. To support above view, we calculated backward trajectories of air parcels arriving at 100 m above ground level over WH every hour between 19:00 and 08:00 UTC, 31 July 2021 using HYSPLIT model (https://www.ready.noaa.gov/HYSPLIT.php) and the Global Forecast System (GFS) reanalysis data (0.25 ° resolution, https://www.emc.ncep.noaa.gov/emc/pages/numerical_forecast_systems/gfs.php). Our intention is not to resolve the dynamical evolution of the MCSs but to analyze atmospheric transport at a relatively larger scale. Figure S3 shows the calculated backward trajectories for WH, together with those for ZB and BZ in the same time window. The trajectories in Figure S3(right) indicate that air parcels influencing the NBL in WH were mostly from the marine boundary layer over the Yellow Sea area. The prevailing wind direction at 850 hPa over the Yellow Sea and the neighbouring land was SW, as shown in Figure 5a in Chen et al. (2022). Such wind condition could facilitate the transport of PPO from the continent to the sea area. Because of the lower emissions of $NO_x$ over the sea, PPO can be well sustained at night and transported to continental locations like WH through sea breezes.

The 24-h backward trajectories for ZB (Figure S3(left)) and BZ (Figure S3(middle)) provide additional clues denying STT impacts on nighttime $O_3$ in these cities. All the trajectories do not indicate any transport of air parcels from altitudes over daytime boundary layer. To gain a more complete insight into the air movements during and before the NOE events, we show in Figure S4 a matrix of backward trajectories for air parcels arriving at 100 m above ground level over the domain 36 °38 'N and 115 °122 'E at 19:00 UTC (03:00 LT), 31 July 2021. The trajectory heights and locations shown in Figure S4 indicate that only 3 of the 24 trajectories travelled over daytime boundary layer and the 3 trajectories ended at locations over the Bohai Gulf. Therefore, our systematic trajectory analysis does not suggest that the NOE events in the NCP cities were related with downward transport of airmasses from the free troposphere.

### 3.4 Confirming rapid downward transport of daytime PPO

To confirm the possibility of rapid downward transport of daytime PPO, we obtained some radiosonde data from three stations in Shandong Province, i.e., Zhangqiu (ZQ, 117.524 'E, 36.713 'N), Rongcheng (RC, 122.477 'E, 37.173 'N) and QD. ZQ is about 52 km east of JN and RC about 56 km southeast of WH. The radiosonde data collected at 19:00 LT, 31 July and 07:00 LT, 1 August 2021 at these sites can be used to get a glimpse of the vertically thermal and dynamical evolutions in the night of 31 July. The raw radiosonde data include temperature, pressure, relative humidity, and wind speed and direction (https://data.cma.cn/). We calculated the virtual temperature and equivalent potential temperature (θse) (Bolton, 1980) and wind shear, $(du/dz)^2$ (Cho et al., 2001). The vertical profiles of these quantities from surface level to 400 hPa are shown in

Figure S5. The pronounced decreasing of θse below 900 hPa over ZQ and QD from 19:00 LT, 31 July to 07:00 LT, 1 August indicated that a descending process occurred at the night. The wind shear peaked near 900 hPa over ZQ and QD, providing kinetic energy for the mixing process. The above thermal and dynamical conditions were favorable for the downward mixing of higher levels of $O_3$ in the residual layer over ZQ and QD. Over RC, however, the thermal and dynamical conditions were different (Figure S5c) and less favorable for triggering the downward transport. This is consistent with the data from the neighboring coast city WH, showing high surface $O_3$ during the NOE event accompanied with relatively high CO and water vapor (Figure S6f). As a coast city near WH, RC should have been impacted by airmasses from marine boundary layer, as discussed for WH in section 3.3.

### 3.5 Cities impacted by the mesoscale convective systems

Above analyses show that not all cities in the study area were clearly influenced by strong downward transport. The cities strongly impacted by the MCSs should have experienced intensive vertical air motion. Chen et al. (2022) showed in their Figure 8 that the vertical atmospheric activity seemed to be limited near ZB and the main convective zone was in the western Shandong Province. However, this figure presents by far not the whole process of the convective activity because it only show six snapshots of the MSCs observed between 20:00 LT, 31 July and 01:00 LT, 1 August 2021 and does not show the dissipation of the MSCs. The radar reflectivity maps in Figure S7 add additional snapshots to the MSCs. As can be seen in Figure S7b-f, ZB and JN were clearly under the influence of the MSCs around 00:00 LT, hit heavily by the MSCs between 01:00 LT and 03:00 LT, 1 August. Without doubt, the MSCs impacting the cities with NOEs reported in Chen et al. (2022) also impacted ZB. The radar reflectivity maps show clearly that except QD and WH, all cities listed in our Table 1 were strongly impacted by the MSCs. While the occurrence of the NOE was a little later than the MSCs impact, the sequence of the MSCs impacts is consistent with that of the NOE events. Therefore, the major conclusions based on our analysis of data from the ZB supersite should also apply to Chen et al. (2022). In next sections, we offer more evidence from ZB to support our argument.

### 3.6 Evidence from ground-based remote sensing

In addition to the backward trajectories, some ground-based remote sensing data from ZB also support our view. Figure S8 presents time-altitude cross sections of $O_3$, relative humidity (RH), virtual temperature (Tv) and wind speed (WS) observed at the ZB supersite between 18:00 LT, 31 July and 06:00 LT, 1 August 2021. As can been seen in the figure, relatively higher $O_3$ mixing ratios occurred only below about 0.5 km. From the evening to midnight, very strong wind prevailed below 1 km, after which humidity was largely enhanced, it rained and the NOE event occurred. The wind directions below 1 km were southerly before 01:00 LT and turned to northerly by 02:00 LT (Figure S9). The top of the higher $O_3$ layer was partly uplifted during the NOE period. According to the backward trajectories shown in Figure S3(left), the air parcels were from the southwest sector and travelled mostly below 0.5 km above sea level. Both the remote sensing data and the trajectory

analysis provide no evidence of air from the free troposphere. Therefore, it is very likely that the NOE observed from the ground to about 0.5 km was due to advection transport of daytime PPO from the southwest sector rather than SST impacts.

**3.7 Diagnosis based on measurements of pollutants in the surface layer**

To further understand the source characteristic of NOE, the hourly average concentrations of surface PAN and $O_3$ observed at the ZB supersite are displayed in Figure 2. It can be seen that the variations of PAN and $O_3$ were in phase and the concentrations of both gases were well correlated ($r^2$=0.60, p<0.0001, n=283), indicating that the variations of PAN and $O_3$ were driven mainly by chemical and physical processes within the boundary layer. The maximum $O_3$ from 02:00 LT to 05:00 LT on 1 August was 88 $\pm$ 5 ppbv, matching with PAN of 0.44 $\pm$ 0.02 ppbv, which was still significantly higher than

those lowest values in nighttime of other dates. As a secondary photochemical pollutant, PAN is thermally unstable but can be transported over higher altitudes, where it has much longer lifetime due to low temperatures. However, observations over the years on mountain tops, aircrafts and satellites showed that the mixing ratios of PAN in the free troposphere and lower stratosphere of the Northern Hemisphere were normally much lower than 0.5 ppb and mostly lower than 0.3 ppb (Singh et al., 2007; Moore and Remedios, 2010; Roiger et al., 2011; Pandey Deolal et al., 2013; Fadnavis et al., 2014; Kramer et al., 2015).

PAN decomposes rapidly in warm urban air. According to Cox and Roffey (1977), the lifetime of PAN was only 2.7 hours at 25 ℃. In our case, the nighttime temperature at the ZB supersite varied from 33.1 ℃ at 18:00 LT on 31 July to 20.4 ℃ at 06:00 LT on 1 August, 2021, with an average of 26.0 ℃. Under such warm condition, the thermal decomposition lifetimes were in the range of 0.2-1.3 hours based on the hourly observations. If the surface layer had been significantly impacted by air masses from the free troposphere and lower stratosphere, we would have seen much lower levels of PAN during the NOE

instead of the observed 0.4-0.5 ppb (Figure 2). Therefore, given the relative lower concentrations of PAN in the free troposphere and lower stratosphere and the rapid thermal decomposition, it is unlikely that over 0.4 ppb of PAN could be observed in surface air significantly impacted by stratospheric intrusion, not to mention that the PAN values during the NOE were even much higher than the nighttime values before and after the NOE event.

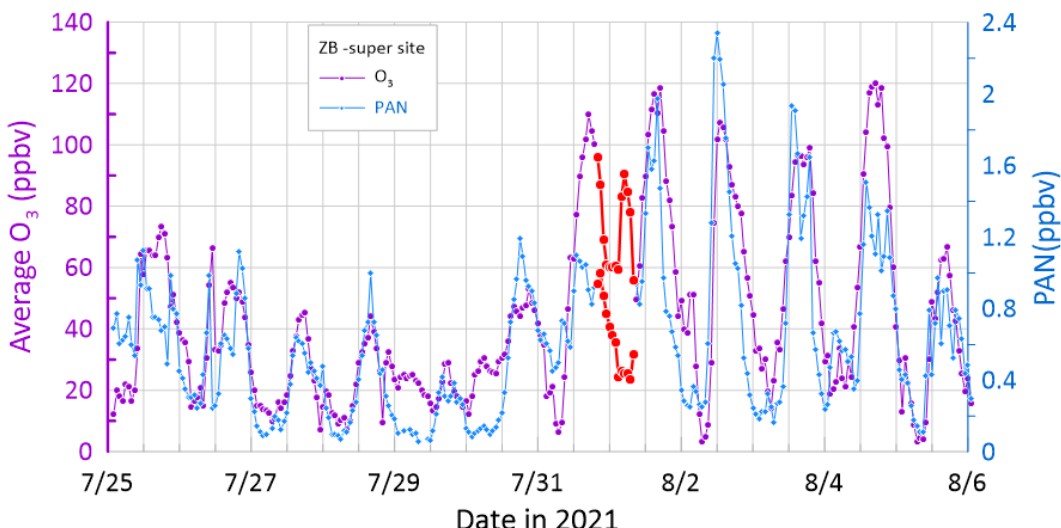

**Figure 2.** Time series of hourly average of $O_3$ (purple) and PAN (blue) at the ZB supersite between 25 July and 5 August 2021. Data from 18:00 LT on 31 July to 06:00 LT on 1 August are highlighted in red.

Photochemical ages of air masses arriving at the ZB supersite from 25 July to 5 August 2021 were estimated based on hourly VOCs (volatile organic compounds) measurements. The methodologies of the estimation are given in detail in Text S2. Figure 3 shows two sets of estimated photochemical ages based on ratios [Butane]/[Propane] and [o-Xylene]/[Ethylbenzene], respectively, together with hourly $O_3$ concentrations from the supersite. As can be seen in the figure, photochemical ages estimated based on [o-Xylene]/[Ethylbenzene] are much shorter than those based on [Butane]/[Propane]. Since we used the observed maximum [Butane]/[Propane] and [o-Xylene]/[Ethylbenzene] in the calculations instead of the respective initial [Butane]/[Propane] and [o-Xylene]/[Ethylbenzene], the photochemical ages were underestimated, particularly those based on measurements of o-xylene and ethylbenzene, which are much more reactive than the alkanes (see Text S2). The estimated photochemical ages can be used to check the major conclusion in Chen et al. (2022) even though they could be underestimated. Here, the actual values of photochemical ages are less important than their variations during, before and after the NOE. The photochemical ages of stratospheric air masses are usually longer than one year (Diallo et al., 2012). However, all estimated photochemical ages in Figure 3, including those for the NOE period, are much shorter than one day. More importantly, the photochemical ages for the NOE from 31 July to 1 August 2021 varied roughly in the middle of all estimated ages and showed no drastic increases, which would be expected if the surface air had contained a significant fraction of stratospheric air. Therefore, our observation-based calculations of photochemical ages do not support the conclusion made by Chen et al. (2022) about the mechanisms of the NOE. It is very likely that the NOE was not a result of typhoon-induced stratospheric intrusion, instead, it was originated from fresh photochemical production of $O_3$ in the lower troposphere.

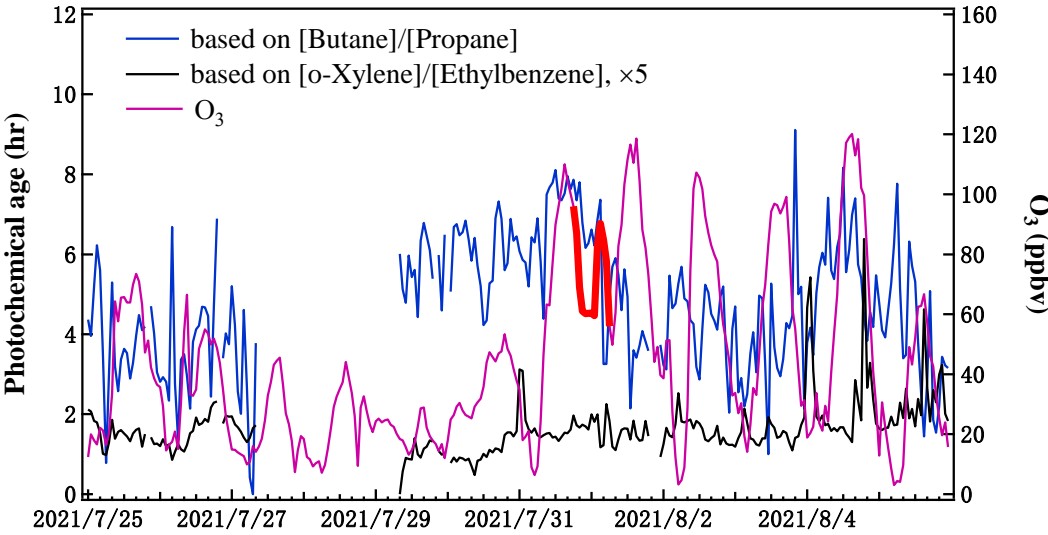

**Figure 3. Variations of estimated photochemical ages of air masses arriving at the ZB supersite and the $O_3$ mixing ratio (purple). The $O_3$ data from 18:00 LT on 31 July to 06:00 LT on 1 August 2021 are highlighted in bold red line. The photochemical ages were estimated on the basis of [Butane]/[Propane] (blue) and [o-Xylene]/[Ethylbenzene] (black), respectively.**

## 4 Discussion and Conclusions

The NOE event presented by Chen et al. (2022) was actually one of the normal cases of NOE associated with the compensatory downdrafts induced by convection cells (Betts et al., 2002). Occurrences of NOE had been reported in the NCP during the summer (e.g., Ma et al., 2013; Jia et al., 2015). Similar phenomena were found in southern China (He et al., 2021), in the Bay of Bengal in India (Sahu and Lal, 2006) and even in Amazonia (Betts et al., 2002). The daytime PPO is transported upward to and resides in the residual layer, while surface $O_3$ is largely removed at night mainly by $NO_x$ titrations,

forming a large positive lower tropospheric gradient of $O_3$ from surface to the residual layer during the night and early morning period, as often reported in the literature (e.g., Ma et al., 2013; Jia et al., 2015; Wang et al., 2017; Tang et al., 2017; Zhao et al., 2019; Zhu et al., 2020). The positive gradient of $O_3$ can be strongly disturbed by nighttime convective processes or low-level jets and the compensatory downdrafts in convection systems can cause NOE events, as reported for a Amazonia site (Betts et al., 2002) and a NCP site (Jia et al., 2015) and systematically summarized in He et al. (2022).

Previous investigations on surface $O_3$ enhancement associated with passage of typhoons revealed two possible mechanisms: i) stratospheric $O_3$ is ultimately transported to the surface level after typhoon-induced STT (Jiang et al., 2015; Wang et al., 2020; Zhan et al., 2020; Chen et al., 2022; Meng et al., 2022); ii) formation and accumulation of $O_3$ as well as emissions of $O_3$ precursors in the boundary layer are promoted under meteorological conditions accompanying strong atmospheric subsidence in typhoon periphery (Hung and Lo, 2015; Shao et al., 2022; Wang et al., 2022). Directly before the

NOE event reported by Chen et al. (2022), the photochemical formation of $O_3$ in the NCP was obviously intensified after a

few days of weakening (Figure 1). The $O_3$-rich air spread within the boundary layer during the daytime of 31 July and remained in the residual layer at night. Given the favorable thermal-dynamical condition like MCSs, the PPO in the residual layer could easily be conveyed downward to the surface, leading to NOE in the surface layer, as also shown in other studies (Shu et al., 2016; Qu et al., 2021; Ouyang et al., 2022; He et al., 2022). Our analysis supports the conclusion that this NOE event was caused by rapid downward transport of daytime PPO residing in the residual layer.

The STT of $O_3$ is often observed in the free troposphere through balloon or aircraft-based observations and air masses associated with identified STT of $O_3$ exhibit usually the feature of HOLWCO. However, air mass with HOLWCO feature in free troposphere do not necessarily mean that the $O_3$ enhancement is originated from the stratosphere (Stoller et al., 1999). For those observations made at a high mountain site (Izaña, 28°18′N, 16°30′W, 2370 m a.s.l), even with stratospheric tracers (such as $^7Be$), the contribution of PPO to the rise of $O_3$ at surface level was important and the stratosphere seemed not to be a direct source (Prospero et al., 1995; Graustein and Turekian, 1996). Although the air with HOLWCO, induced by katabatic winds, was observed at the base camp of Mount Everest (about 5000 m a.s.l.), the source of $O_3$ from the stratosphere was not confirmed (Zhu et al., 2006). Simultaneous observations of $O_3$ and PAN at the Namco (4545 m a.s.l.) in Tibet captured air masses with high $O_3$ and low water vapor, which were accompanied with increases of PAN, suggesting that PPO during the long-range transport might be one of the major sources of elevated $O_3$ (Xu et al., 2018). These examples show that even at high altitude sites, HOLWCO phenomena may not be caused by STT.

In the NOE cases reported in Chen et al. (2022), the HOLWCO feature during the NOE was very much far from the stratospheric characteristics. The maximum $O_3$ levels were around 80 ppb and significantly lower than respective daytime maxima; the CO levels were between 200 and 500 ppb (see Figure 3 in Chen et al., 2022 and Figure S6), much higher than the CO levels in the middle and upper troposphere (about 100 ppb) and lower stratosphere (<50 ppb) (Inness et al., 2022) and even not lower than those CO values on some other days; the measured water vapor pressure during the NOE was close to its normal values (Figure S6) and did not show any sign of substantial stratospheric impact. In other words, although the levels of CO or water vapor were relatively lower during the NOE events, they did not show large deviations from their values within normal boundary layer, nor substantial STT influences. All these, together with our evidence given above, indicate that the NOE reported in Chen et al. (2022) was not caused by typhoon-induced stratospheric intrusion. Therefore, more cautions should be taken when attributing HOLWCO events at low altitude sites to stratospheric impact, whether or not there was an influence from a typhoon or other synoptic system. It is suggested that if possible, each case should be verified by analyzing both physical and chemical processes before making a conclusion.

**Data availability.** The data used in this study are available upon request to the corresponding author (xiaobin_xu@189.cn).

**Author contributions.** XZ and XX designed the research. WY was responsible for the observations at the Zibo supersite. CG and YL validated the data from the supersite. XZ, YS and XX performed the data analysis. XZ and XX prepared the manuscript.

**Competing interests.** The contact author has declared that none of the authors has any competing interests.

**Financial support.** This research has been supported by the National Natural Science Foundation of China (grant no.
41775031) and the Science and Technology Development Fund of the Chinese Academy of Meteorological Sciences (grant nos. 2023KJ013 and 2023KJ014). The observations at the ZB supersite were supported by the National Research Program for Key Issues in Air Pollution Control (No. DQGG202119, DQGG202137).

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
