# Peer review of "Comment on “Transport of substantial stratospheric ozone to the surface by a dying typhoon and shallow convection” by Chen et al. (2022)"

_EGUsphere, 2023_

## Referee Comment (RC1)

This manuscript comments on the study of Chen et al. (2022), proposing that the nocturnal O3 enhancement (NOE) on 31 July 2021 over the North China Plain is caused by the stratosphere O3 intrusion after the passage of Typhoon In-fa. The authors argue that the NOE is instead due to the photochemically produced O3, which can be brought downward from the residual layer at night. Overall, the manuscript is well-written, and the method is reasonable. I have a few points that I think could be addressed to strengthen the manuscript and some minor comments.

General Comments:

1. Although I tend to believe that STT usually has little effect on ground O3 at low altitudes, the reasoning in the manuscript needs to be further elaborated. Could you explain more on why higher nighttime Ox than daytime O3 is an indicator of downward mixing from the residual layer and vice versa? Also, both O3 and PAN can be transported over long distances, and I am not sure why the in-phase changes of O3 and PAN could exclude the possibility of STT. Lastly, the authors mention that HOLWCO could also occur in photochemically aged air, not necessarily from STT, but the photochemical age analysis in the manuscript shows fresh air with ages less than 10 hrs. Does that mean HOLWCO can occur in fresh air as well?

2. If possible, the vertical profiles of virtual temperature and winds can be added to further support the conclusion that the rapid downward transport of daytime PPO.

Minor Comments:

Line 43: 'rangeby' should be 'range by'.

Line 66-67: Better to add how many sites there are in each city to the main texts and mention that Fig. 1 is the site average.

Line 87-90: Consider putting an enlarged figure in the supplement and mark these NOEs. It is hard to identify in Figure 1.

Line 127: Change 'arrived' to 'arriving'.

The caption of Figure 3: Change 'arrived' to 'arriving' and 'based' to 'based on'.

---

## Author Comment (AC1)

**We are grateful to the reviewers for their comments and suggestions. Here are our responses (in blue) to reviewers' comments.**

**Xiaobin Xu on behalf of all co-authors**

**Response to Reviewer #1:**

This manuscript comments on the study of Chen et al. (2022), proposing that the nocturnal $O_3$ enhancement (NOE) on 31 July 2021 over the North China Plain is caused by the stratosphere $O_3$ intrusion after the passage of Typhoon In-fa. The authors argue that the NOE is instead due to the photochemically produced $O_3$, which can be brought downward from the residual layer at night. Overall, the manuscript is well-written, and the method is reasonable. I have a few points that I think could be addressed to strengthen the manuscript and some minor comments.

**Response: We thank the reviewer for the positive comments and suggestions. Here are our point-to-point responses to the comments.**

General Comments:

1. Although I tend to believe that STT usually has little effect on ground $O_3$ at low altitudes, the reasoning in the manuscript needs to be further elaborated. Could you explain more on why higher nighttime Ox than daytime $O_3$ is an indicator of downward mixing from the residual layer and vice versa? Also, both $O_3$ and PAN can be transported over long distances, and I am not sure why the in-phase changes of $O_3$ and PAN could exclude the possibility of STT. Lastly, the authors mention that HOLWCO could also occur in photochemically aged air, not necessarily from STT, but the photochemical age analysis in the manuscript shows fresh air with ages less than 10 hrs. Does that mean HOLWCO can occur in fresh air as well?

● Could you explain more on why higher nighttime Ox than daytime O3 is an indicator of downward mixing from the residual layer and vice versa?

**Response: Our intention is to check whether or not surface $O_3$ observed during the NOE contained significant contribution of $O_3$ from the stratosphere. For this purpose, a comparison of $O_3$ levels in the vertical direction is helpful. However, $O_3$ is very reactive and can be significantly removed by titration reactions in the boundary layer, with $O_3 + NO = NO_2 + O_2$ being the most important one. Therefore, $O_x$ ($O_3+NO_2$) is a more conserved quantity than $O_3$ and hence a better metric for comparison (Kley et al., 1994; Kleinmann et al., 2002; Caputi et al., 2019; He et al., 2022). During daytime, $NO_2$ formed in the titration reaction is rapidly photolyzed to regenerate $O_3$ so that the net chemical loss of $O_3$ is relatively small. At night, however, the reaction between $O_3$ and NO leads to lower levels of $O_3$ in the**

nocturnal boundary layer (NBL) and in the residual layer (RL). Because the emission of NO takes place mainly in the NBL, much more $O_3$ is removed by the titration reaction in the NBL than in the RL (Wang et al., 2018; Caputi et al., 2019; He et al., 2022). In addition, $O_3$ in the NBL is subjected to dry deposition. Therefore, the $O_3$ level before sunset largely remains in the RL (Caputi et al., 2019; He et al., 2022) and is usually much higher in the RL than in the NBL under normal conditions.

Following the method of He et al. (2022), we make comparison of afternoon $O_3$ averages on 31 July with the respective $O_x$ averages in the NBL during the NOE between 31 July and 1 August, 2021. To facilitate the comparison, we treat the average surface $O_3$ during 14:00-17:00 LT of 31 July as afternoon average of $O_3$ in the convective boundary layer, denoted as $[O_3]_{aft}$. Let us now focus on three nighttime atmospheric conditions, (I) undisturbed, (II) disturbed with NOE but no STT impact, and (III) disturbed with NOE and significant STT impact.

Under undisturbed condition (I), the nighttime average $O_3$ concentration in the RL ($[O_3]_{RL}$) should be close to (or only slightly lower than) $[O_3]_{aft}$ (Caputi et al., 2019; He et al., 2022), while the average $O_3$ concentration in the NBL ($[O_3]_{NBL}$) should be much lower than $[O_3]_{aft}$ due to the impacts of NO titration ($\Delta[O_3]_{titr}$) and dry deposition ($\Delta[O_3]_{dep}$), and the average $O_x$ concentration in the NBL ($[O_x]_{NBL}$) should also be lower than $[O_3]_{aft}$ due to dry deposition. The following relationships should be tenable:

$$[O_3]_{RL} \leq [O_3]_{aft} \tag{1}$$
$$[O_3]_{NBL} = [O_3]_{aft} - \Delta[O_3]_{titr} - \Delta[O_3]_{dep} \tag{2}$$
$$[O_x]_{NBL} = [O_3]_{aft} - \Delta[O_3]_{dep} \tag{3}$$

Under disturbed condition with NOE but no STT impact (II), a downward transport of $O_3$ from the RL to NBL should be considered. Assuming that the downward transport causes a reduction of $[O_3]_{RL}$ by $\Delta[O_3]_{D1}$ and an increase of $[O_3]_{NBL}$ by $\Delta[O_3]_{D2}$, then

$$[O_3]_{RL} \leq [O_3]_{aft} - \Delta[O_3]_{D1} \tag{4}$$
$$[O_3]_{NBL} = [O_3]_{aft} - \Delta[O_3]_{titr} - \Delta[O_3]_{dep} + \Delta[O_3]_{D2} \tag{5}$$
$$[O_x]_{NBL} = [O_3]_{aft} - \Delta[O_3]_{dep} + \Delta[O_3]_{D2} \tag{6}$$

Under disturbed condition with NOE and STT impact (III), net contributions of $O_3$ from the STT should be considered to the RL and the NBL. Assuming that the STT contribution increases $[O_3]_{RL}$ and $[O_3]_{NBL}$ by $\Delta[O_3]_{STT1}$ and $\Delta[O_3]_{STT2}$, respectively, then

$$[O_3]_{RL} \leq [O_3]_{aft} + \Delta[O_3]_{STT1} \tag{7}$$
$$[O_3]_{NBL} = [O_3]_{aft} - \Delta[O_3]_{titr} - \Delta[O_3]_{dep} + \Delta[O_3]_{STT2} \tag{8}$$
$$[O_x]_{NBL} = [O_3]_{aft} - \Delta[O_3]_{dep} + \Delta[O_3]_{STT2} \tag{9}$$

Equation (3) indicates that $[O_x]_{NBL}$ should be significantly lower than $[O_3]_{aft}$ under

undisturbed conditions. Although equation (6) shows that $[O_x]_{NBL}$ could be higher than $[O_3]_{aft}$ (i.e., if $\Delta[O_3]_{dep} < \Delta[O_3]_{D2}$), it cannot really occur because $[O_3]_{RL} < [O_3]_{aft}$ (see equation (4)) and $O_3$ cannot be transported from a lower concentration position to the higher one. Therefore, $[O_x]_{NBL}$ should not be significantly higher than $[O_3]_{aft}$ under disturbed conditions with NOE but no STT impact. Dry deposition is only a small sink for nighttime surface $O_3$ in northern China (Tang et al., 2017), while a STT impact could substantially enhanced the level of surface $O_3$ if it reaches the surface layer. Hence, it is very likely according to equation (9) that $[O_x]_{NBL}$ is significantly higher than $[O_3]_{aft}$ under disturbed condition with a STT impact. In summary, $[O_x]_{NBL}$ should be significantly higher than $[O_3]_{aft}$ if the NBL is really impacted by stratospheric $O_3$, otherwise the STT impact is negligible even though a NOE event is observed.

As shown in Table 1 in our manuscript, in all cities but Weihai (WH), mean $O_x$ during the maximum NOE period was either much lower than or nearly equal to mean $O_3$ during the afternoon period. This indicates that $O_3$ in the NBL in those cities was probably not impacted by stratospheric $O_3$. Mean $O_x$ in WH during the NOE was much higher than mean $O_3$ in the erstwhile afternoon. However,this could not be attributed to stratospheric impact. Chen et al. (2022) mentioned "the surface ozone enhancement at midnight was not coincident with CO reduction, suggesting that the stratospheric air mass might not have reached the surface." (at the end of the second paragraph in section 4.2). The higher $[O_x]_{NBL}$ in WH may also be interpreted by the above equation (6), i.e., if $\Delta[O_3]_{D2}$ was much greater than $\Delta[O_3]_{dep}$. A great $\Delta[O_3]_{D2}$ would require a high $[O_3]_{aft}$, either from local photochemical production or from transport. Actually, photochemically produced $O_3$ (PPO) had not been well established in WH during the daytime on 31 July, as can be seen in our Figure 1f. Therefore, regional-scale transport of PPO is suspected. In section 3 of our manuscript, we provided an interpretation in the viewpoint of diurnal alternations of land-sea breezes. Our view is supported by the backward trajectories shown in Figure 1R(right), which indicate that air parcels influenced the NBL in WH were mostly from the Yellow Sea area. The prevailing wind direction at 850 hPa over the Yellow Sea and the neighboring land was SW, as shown in Figure 5 (a) in Chen et al. (2022). Such wind condition could facilitate the transport of PPO from the continent to the sea area. Because of the lower emissions of $NO_x$ over the sea, PPO can be well sustained at night and transported to continental locations like WH through sea breezes.

In Figure R1, we also show 24-h backward trajectories arriving at 100 m above ground over ZB and BZ. All the trajectories do not indicate any transport of air parcels from altitudes over daytime boundary layer. Figure R2 shows a matrix of backward trajectories for air parcels arriving at 100 m above ground over the

domain 36°-38°N and 115°-122°E at 19:00 UTC, 31 July 2021. Only 3 of the 24 trajectories were ever travelled over daytime boundary layer and the 3 trajectories ended at locations over the Bohai Gulf.

[Figure]

**Figure R1. Backward trajectories for air parcels arriving at 100 m above ground over Zibo (ZB, left), Binzhou (BZ, middle) and Weihai (WH, right) every hour between 19:00 and 08:00 UTC, 31 July 2021. The trajectories were computed online using the HYSPLIT model (https://www.ready.noaa.gov/HYSPLIT.php; Stein et al. 2015; Rolph et al., 2017) and the Global Forecast System (GFS) reanalysis data (0.25° resolution, https://www. emc.ncep. noaa.gov/emc/pages/numerical_forecast_systems/gfs.php). The total run time for the trajectories was 24 hours.**

[Figure]

**Figure R2. Matrix of backward trajectories for air parcels arriving at 100 m above ground over the domain 36°-38°N and 115°-122°E at 19:00 UTC, 31 July 2021. The trajectories were computed online using the HYSPLIT model (https://www.ready.noaa.gov/HYSPLIT.php; Stein et al. 2015; Rolph et al., 2017) and the Global Forecast System (GFS) reanalysis data (0.25° resolution, https://www.emc.ncep.noaa.gov/emc/pages/numerical_forecast_systems/gfs. php). The total run time for the trajectories was 24 hours.**

In addition to the backward trajectories, some ground-based remote sensing data also support our view. Figure R3 presents vertical profiles of $O_3$, relative humidity, virtual temperature and wind speed observed at the ZB supersite between 18:00 LT, 31 July and 06:00 LT, 1 August 2021. As can been seen in the figure, relatively higher $O_3$ mixing ratios occurred only below about 0.5 km. From the evening to midnight, very strong wind prevailed below 1 km, after which humidity was largely enhanced, it rained and the NOE event occurred. The top of the higher $O_3$ layer was partly uplifted during the NOE period. According to the backward trajectories shown in Figure R2 (left), the air parcels were from the southwest sector and travelled mostly below 0.5 km above sea level. There was no indication of air from the free troposphere. And it is very likely that the NOE was due to horizontal transport of daytime PPO from the southwest sector.

[Figure]

Figure R3. Vertical profiles of $O_3$, relative humidity (RH), virtual temperature (Tv) and wind speed (WS) observed at the ZB supersite between 18:00 LT, 31 July and 06:00 LT, 1 August 2021.

In conclusion, our comparisons of nighttime $O_x$ with daytime $O_3$ in the NCP cities and analysis of backward trajectories and remote sensing data suggest that the NOE events observed at the NCP cities were most likely from the transport of $O_3$ in the RL or (in the case of WH) in the marine boundary layer, and unlikely from the transport of stratospheric $O_3$.

● Also, both $O_3$ and PAN can be transported over long distances, and I am not sure why the in-phase changes of $O_3$ and PAN could exclude the possibility of STT.

Response: Indeed, both $O_3$ and PAN can be transported over long distances at higher altitudes, where their chemical removal is limited. However, PAN is primarily produced where VOCs and $NO_x$ are largely emitted, such as in industrial and populated areas or in biomass burning plumes (Xu et al., 2018). Higher temperatures and intense radiation are favorable meteorological conditions for the formation of PAN. These chemical and meteorological conditions facilitate the production of $O_3$ as well. For this reason, $O_3$ and PAN in the boundary layer are usually well correlated, particularly in warm seasons. The in-phase changes of $O_3$ and PAN reflect the dominant influences of chemical and physical processes within the boundary layer but do not exclude the possibility of STT. What really do not support the possibility of STT is the relatively higher value of PAN observed during the NOE at the ZB supersite. The average PAN (0.44±0.02 ppbv) was higher than the lowest nighttime PAN levels on most days before and after the NOE event. This evidence disproves the possibility of significant STT impact.

Lacking of parallel measurements of PAN at high altitudes over the NCP hinders us from making a vertical comparison of PAN levels. However, some PAN values from earlier high altitude observations can be referenced. The PAN values in the UTLS over the NCP in August 2003 ranged from about 0.17 ppb (at 185 hPa) to 0.35 ppb (at 278 hPa), as retrieved from the satellite observations (Moore and Remedios, 2010). Ten-years of satellite observations showed that average PAN in the UTLS over the anticyclone region (60–120 °E, 10–40 °N) reached its maximum in summer and varied in the range of 0.06-0.3 ppb (Fadnavis et al., 2014). Aircraft measurements of PAN in July 2008 over Longyearbyen, Spitsbergen (78 °130′N, 15 °330′E) and Oberpfaffenhofen, Germany (48 °40′N, 11 °160′E) showed PAN levels in the range of 0.05-0.3 ppb in the free troposphere and lower stratosphere (Roiger et al., 2011). Airborne measurements over North America showed summer PAN in the free troposphere and lowermost stratosphere were usually lower than 0.5 ppb unless biomass burning air masses were sampled (Singh et al., 2007). Observations at high mountain sites also indicated quite low PAN levels in the free troposphere. For example, multiyear observations at the high alpine site Jungfraujoch (46.55 °N, 7.98 °E, 3580 ma.s.l) found average summer PAN levels of lower than 0.2 ppb in free

tropospheric air masses (Pandey Deolal et al., 2013) and those at Summit station, Greenland (72.34 °N, 38.29 °W, 3212 ma.s.l.) showed average summer PAN levels of lower than 0.1 ppb (Kramer et al., 2015). In summary, the mixing ratios of PAN in the free troposphere and lower stratosphere of the Northern Hemisphere are normally much lower than 0.5 ppb and mostly lower than 0.3 ppb.

PAN is thermally unstable and decomposes rapidly in warm urban air. According to Cox and Roffey (1977), the lifetime of PAN was only 2.7 hours at 25 °C. In our case, the nighttime temperature at the ZB supersite varied from 33.1 °C at 18:00 LT on 31 July to 20.4 °C at 06:00 LT on 1 August, 2021, with an average of 26.0 °C. Under such warm condition, even shorter lifetimes of PAN are expected. The thermal decomposition lifetimes were in the range of 0.2-1.3 hours based on the hourly observations. If the surface layer had been significantly impacted by air masses from the free troposphere and lower stratosphere, we would have seen much lower levels of PAN during the NOE instead of the observed 0.4-0.5 ppb (Figure 2). Therefore, given the relative lower concentrations of PAN in the free troposphere and lower stratosphere and the rapid thermal decomposition, it is unlikely that over 0.4 ppb of PAN could be observed in surface air significantly impacted by stratosphere intrusion, not to mention that the PAN values during the NOE were even much higher than the nighttime values of other dates.

- Lastly, the authors mention that HOLWCO could also occur in photochemically aged air, not necessarily from STT, but the photochemical age analysis in the manuscript shows fresh air with ages less than 10 hrs. Does that mean HOLWCO can occur in fresh air as well?

Response: Chen et al. (2022) attributed the ozone-rich and CO-poor phenomenon during the NOE to the downward transport of stratospheric air. In our manuscript we mentioned "photochemically aged pollution air masses may also show HOLWCO features as observed in the PEM-West A campaign (Newell et al. 1996b; Stoller et al., 1999)" and "air mass with HOLWCO feature in free troposphere do not necessarily mean that the $O_3$ enhancement is originated from the stratosphere (Stoller et al., 1999)". Since HOLWCO phenomena may not be caused by STT, we suggest that more cautions should be taken when attributing HOLWCO events at low altitude sites to stratospheric impact.

The reviewer is asking the possibility of HOLWCO occurring in fresh air, referring the relatively shorter lifetimes we estimated. Our results can neither confirm nor exclude this possibility. Our estimates of photochemical age indicate that surface air at ZB during the NOE was nearly as fresh as at other nights, against a significant impact of stratospheric air, which has ages of years. In addition, the HOLWCO feature during the NOE was very much far from the stratospheric

characteristics. The maximum $O_3$ levels were around 80 ppb and significantly lower than respective daytime maxima; the CO levels were between 200 and 500 ppb (see Figure 3 in Chen et al., 2022 and Figure R4), much higher than the CO levels in the middle and upper troposphere (about 100 ppb) and lower stratosphere (<50 ppb) (Inness et al., 2022) and even not lower than those CO values on some other days; the measured water vapor pressure during the NOE was close to its normal values (Figure R4) and did not show any sign of substantial stratospheric impact. In other words, although the levels of CO or water vapor were relatively lower during the NOE event, they did not show deviations from their values within normal boundary layer, nor substantial STT influences. All these, together with our evidence given above, indicate that the NOE reported in Chen et al. (2022) was not caused by typhoon-induced stratospheric intrusion.

[Figure]

Figure R4. Time series of hourly multi-sites-averages of $O_3$ (purple), CO (black ) and water vapor pressure (blue) between 25 July and 5 August 2021. Data from 18:00 LT on 31 July to 06:00 LT on 1 August are highlighted in thick lines. The multi-site data from HS (4 sites) (a), BZ (6 sites) (b), JN (13 sites)(c), WF (6 sites) (d), QD (13 sites) (e), WH (5 sites)(f) and ZB (7sites) (g) are available at https://quotsoft.net/air (last access: 15 April 2023; X. L. Wang, 2020). Data from the ZB supersite (h) is provided by the Chinese Academy of Environmental Sciences (CRAES). Water vapor pressure data are from https://data.cma.cn/.

[Figure]

**Figure R4. (continued).**

2. If possible, the vertical profiles of virtual temperature and winds can be added to further support the conclusion that the rapid downward transport of daytime PPO.

**Response. There are three routine radiosonde stations in the studied area, i.e., Zhangqiu (ZQ, 117.524 ˚E, 36.713 ˚N), Rongcheng (RC, 122.477 ˚E, 37.173 ˚N) and QD. ZQ is about 52 km east of JN and RC about 56 km southeast of WH. The radiosonde data collected at 19:00 LT, 31 July and 07:00 LT, 1 August 2021 at these sites can be used to get a glimpse of the vertically thermal and dynamical evolutions in the night of 31 July. The raw radiosonde data include temperature, pressure, relative humidity, and wind speed and direction (https://data.cma.cn/). We calculated the virtual temperature and equivalent potential temperature($\theta_{se}$) (Bolton, 1980) and wind shear, $(du/dz)^2$ (Cho et al., 2001). These vertical profiles of these quantities from surface level to 400 hPa are shown in Figure R5. The pronounced decreasing of $\theta_{se}$ below 900 hPa over ZQ and QD from 19:00 LT, 31 July to 07:00 LT, 1 August indicated a descending process occurred at the night. The wind shear peaked near 900 hPa over ZQ and QD, providing kinetic energy for the descending and mixing**

**process. The above thermal and dynamical conditions were, of course, favorable for the downward mixing of higher levels of O₃ in the residual layer over ZQ and QD. Over RC, however, the thermal and dynamical conditions were different (Figure R5c) and less favorable for triggering the downward transport. This is consistent with the data from the neighboring coast city WH, showing high surface O₃ during the NOE event accompanied with relatively high CO and water vapor (Figure R4f).**

[Figure]

**Figure R5. The vertical profiles of virtual temperature (red), equivalent potential temperature (θse, grey) and wind shear (blue) calculated on the basis of routine radiosonde observations data at 19:00 LT, 31 July and 07:00 LT, 1 August 2021 in Zhangqiu (ZQ)(a), Qingdao (QD )(b) and Rongcheng (RC) (c).**

**We will include supporting materials and key points of above discussion in our revision.**

Minor Comments:

Line 43: 'rangeby' should be 'range by'.

**Response: Thank you. This error will be corrected.**

Line 66-67: Better to add how many sites there are in each city to the main texts and mention that Fig. 1 is the site average.

**Response: Yes, the site numbers will be added in the revised manuscript.**

Line 87-90: Consider putting an enlarged figure in the supplement and mark these NOEs. It is hard to identify in Figure 1.

**Response : Yes, a figure will be included in the revised supplement to show an enlarged view of the NOEs.**

Line 127: Change 'arrived' to 'arriving'.

The caption of Figure 3: Change 'arrived' to 'arriving' and 'based' to 'based on'.

**Response : Will be changed.**

**Reference**

**Bolton, D .: The computation of equivalent potential temperature, Mon. Weather Rev., 108, 1046-1053, https://doi.org/10.1175/1520-0493(1980)108<1046: TCOEPT>2.0.CO;2, 1980.**

**Caputi, D.J., Faloona, I., Trousdell, J., Smoot, J., Falk, N., and Conley, S.: Residual layer ozone, mixing, and the nocturnal jet in California's San Joaquin Valley, Atmos. Chem. Phys., 19, 4721–4740, https://doi.org/10.5194/acp-19-4721-2019, 2019.**

**Chen, Z., Liu, J., Qie, X., Cheng, X., Shen, Y., Yang, M., Jiang, R., and Liu., X.: Transport of substantial stratospheric ozone to the surface by a dying typhoon and shallow convection, Atmos. Chem. Phys., 22, 8221–8240, https://doi.org/10.5194/acp-22-8221-2022, 2022.**

**Cho, J.Y.N., Newell, R.E., Browell, E.V. , Grant, W.B., Butler, C.F., and Fenn, M.A.: Observation of pollution plume capping by a tropopause fold, Geophys. Res. Lett., 28(17), 3243-3246, https://doi.org/10.1029/2001GL012898, 2001.**

**Cox, R.A. and Roffey, M.J.: Thermal decomposition of peroxyacetylnitrate in the presence of nitric oxide. Environ. Sci. Technol., 11, 900–906, 1977.**

**Fadnavis, S., Schultz, M.G., Semeniuk, K., Mahajan, A.S., Pozzoli, L., Sonbawne, S., Ghude, S.D., Kiefer, M., and Eckert, E.: Trends in peroxyacetyl nitrate (PAN) in the upper troposphere and lower stratosphere over southern Asia during the summer monsoon season: regional impacts, Atmos. Chem. Phys., 14, 12725–12743, https://doi.org/10.5194/acp-14-12725-2014, 2014.**

**He, C., Lu, X., Wang, H., Wang, H., Li, Y., He, G., He, Y., Wang, Y., Zhang, Y., Liu, Y., Fan, Q., and Fan, S.: The unexpected high frequency of nocturnal surface ozone enhancement events over China: characteristics and mechanisms, Atmos. Chem. Phys., 22, 15243–15261, https://doi.org/10.5194/acp-22-15243-2022, 2022.**

**Inness, A., Aben, I., Ades, M., Borsdorff, T., Flemming, J., Jones, L., Landgraf, J., Langerock, B., Nedelec, P., Parrington, M., and Ribas, R.: Assimilation of S5P/TROPOMI**

carbon monoxide data with the global CAMS near-real-time system, Atmos. Chem. Phys., 22, 14355–14376, https://doi.org/10.5194/acp-22-14355-2022, 2022.

Kleinman, L., Daum, P., Lee, Y.-N., Nunnermacker, L., Springston, S., Weinstein-Lloyd, J., and Rudolph, J.: Ozone production efficiency in an urban area, J. Geophys. Res., 107, 4733, https://doi.org/10.1029/2002JD002529, 2002.

Kley, D., Geiss, H., Mohnen, V.A.: Tropospheric ozone at elevated sites and precursor emissions in the United States and Europe, Atmos. Environ., 28, 149-158, 1994.

Kramer, L.J., Helmig, D., Burkhart, J.F., Stohl, A., Oltmans, S., and Honrath, R.E.: Seasonal variability of atmospheric nitrogen oxides and non-methane hydrocarbons at the GEOSummit station, Greenland, Atmos. Chem. Phys., 15, 6827–6849, https://doi.org/10.5194/acp-15-6827-2015, 2015.

Pandey Deolal, S., Staehelin, J., Brunner, D., Cui, J., Steinbacher, M., Zellweger, C., Henne, S., and Vollmer, M.K.: Transport of PAN and NOy from different source regions to the Swiss high alpine site Jungfraujoch, Atmos. Envron., 64, 103–115, https://doi.org/10.1016/j.atmosenv.2012.08.021, 2013.

Roiger, A., Aufmhoff, H., Stock, P., Arnold, F., and Schlager, H.: An aircraft-borne chemical ionization – ion trap mass spectrometer (CI-ITMS) for fast PAN and PPN measurements, Atmos. Meas. Tech., 4, 173–188, https://doi.org/10.5194/amt-4-173-2011, 2011.

Rolph, G., Stein, A., and Stunder, B.: Real-time Environmental Applications and Display sYstem: READY. Environmental Modelling & Software, 95, 210-228, https://doi.org/10.1016/j.envsoft.2017.06.025, 2017.

Singh, H.B., Salas, L., Herlth, D., Kolyer, R., Czech, E., Avery, M., Crawford, J.H., Pierce, R.B., Sachse, G.W., Blake, D.R., Cohen, R.C., Bertram, T.H., Perring, A., Wooldridge, P.J., Dibb, J., Huey, G., Hudman, R.C., Turquety, S., Emmons, L.K., Flocke, F., Tang, Y., Carmichael, G.R., and Horowitz, L.W.: Reactive nitrogen distribution and partitioning in the North American troposphere and lowermost stratosphere, J. Geophys. Res., 112, D12S04, https://doi.org/10.1029/2006JD007664, 2007.

Stein, A.F., Draxler, R.R, Rolph, G.D., Stunder, B.J.B., Cohen, M.D., and Ngan, F.: NOAA's HYSPLIT atmospheric transport and dispersion modeling system, Bull. Amer. Meteor. Soc., 96, 2059-2077, http://dx.doi.org/10.1175/BAMS-D-14-00110.1., 2015.

Tang, G., Zhu, X., Xin, J., Hu, B., Song, T., Sun, Y., Wang, L., Wu, F., Sun, J., Cheng, M., Chao, N., Li, X., Wang, Y.: Modelling study of boundary-layer ozone over northern China - Part II: Responses to emission reductions during the Beijing Olympics, Atmos. Res., 193, 83–93, http://dx.doi.org/10.1016/j.atmosres.2017.02.014, 2017.

Wang, H., Lu, K., Chen, X., Zhu, Q., Wu, Z., Wu, Y., and Sun, K.: Fast particulate nitrate formation via N2O5 uptake aloft in winter in Beijing, Atmos. Chem. Phys., 18, 10483–10495, https://doi.org/10.5194/acp-18-10483-2018, 2018.

Xu, X., Zhang, H., Lin, W., Wang, Y., Xu, W., and Jia, S.: First simultaneous measurements of peroxyacetyl nitrate (PAN) and ozone at Nam Co in the central Tibetan Plateau: impacts from the PBL evolution and transport processes, Atmos. Chem. Phys., 18, 5199–5217, https://doi.org/10.5194/acp-18-5199-2018, 2018.

**Response to Reviewer #2:**

The study from Zheng et al. is a comment on the study by Chen et al. (2022). Chen et al. (2022) demonstrated that the intrusion of stratosphere ozone induced by Typhoon Infa served as a potential source of surface ozone, and the following shallow local mesoscale convective systems facilitated the downward transport of this potential ozone source to the surface, and led to a nocturnal ozone enhancement (NOE) event in the North China Plain (NCP). Zheng et al. analyzed observations (including PAN, VOCs) from the ZiBo supersite at eastern NCP, and argued that the NOE event originated from fresh ozone photochemical production in the lower troposphere rather than from the stratosphere as pointed out by Chen et al. (2022). This is supported by 1) comparable nighttime surface $O_x$ and daytime $O_3$ levels, 2) a strong correlation between PAN and ozone during the NOE event, and 3) the short photochemical ages of air mass. Overall, the study provides strong observational evidence on the conclusion, and is well-designed and well-written. I suggest some revisions before publication.

**Response: We thank the reviewer for the positive comments.**

While I agree with Zheng et al. that the mixing of ozone produced photochemically and stored in the residual layer has a large contribution to the Zibo NOE event, it does not necessarily exclude the possibility of stratospheric contribution proposed by Chen et al. (2022). Zheng et al. mainly used observations at ZiBo city, however, as seen from Figure 8 of Chen et al. (2022), there is limited vertical atmospheric activity near the ZiBo supersite, as the main convective zone is located in the western Shandong Province. So the difference in the study region between Chen et al. (2022) and Zheng et al. may also contribute to the different conclusions. I would suggest the authors consider this point in their analysis.

**Response: Indeed, Figure 8 of Chen et al. (2022) indicates that the vertical atmospheric activity seemed to be limited near Zibo and the main convective zone was in the western Shandong Province. However, this figure presents by far not the whole process of the convective activity because it only show six snapshots of the mesoscale convective system (MSC) observed between 20:00 LT, 31 July and 01:00 LT, 1 August 2021 and does not show the dissipation of the MSC. The radar reflectivity maps in Figure R6 add additional snapshots to the MSC. As can be seen in Figures R6b-R6f, Zibo and Jinan were clearly under the influence of the MSC around 00:00 LT, hited heavily by the MSC between 01:00 LT and 03:00 LT, 1 August. Without doubt, the MSC impacting the cities with NOEs reported in Chen et al. (2022) also impacted Zibo. The radar reflectivity maps show clearly that except QD and WH, all cities listed in our Table 1 were strongly impacted by the MSC. While the occurrence of the NOE was a little later than the MSC impact, the sequence of the MSC impacts is consistent with that of the NOE events. Therefore, the major conclusions based on our analysis apply also to Chen et al. (2022).**

[Figure]

**Figure R6. Radar reflectivity maps for the North China Plain at (a) 15:06 UTC (23:00 LT, 31 July), (b) 16:12 UTC (00:12 LT, 1 August), (c) 17:00 UTC (01:00 LT, 1 August), (d) 18:06 UTC (02:06 LT, 1 August), (e) 19:00 UTC (03:00 LT, 1 August) and (f ) 20:00 UTC (04:00 LT, 1 August). Some of the major cities are indicated on the maps. The white star on each map shows the location of Zibo.**

One small comment is to clarify $O_3$ as "daytime $O_3$" and Ox as "nighttime Ox" in Table 1
**Response: Thank you. We will added clarifications in the revised manuscript.**

---

## Author Response (AR1)

**Dear Editor,**

**Thank you for handling our manuscript. We are grateful to the reviewers for their comments and suggestions. Originally, we focused our analysis and discussion mainly on chemical processes. In response to the reviewers' comments, we extended our analysis to dynamical processes using backward trajectories, radiosonde and ground-based remote sensing data. We revised our manuscript and supplement by including the results and discussion of the additional analysis. We made a structural change by dividing the contents in section 3 into several subsections. We also made some other changes necessary and corrected error we identified. Here are our responses (in blue) to reviewers' comments and changes (highlighted in yellow) in manuscript and supplement.**
**Xiaobin Xu on behalf of all co-authors**

**Response to Reviewer #1:**

This manuscript comments on the study of Chen et al. (2022), proposing that the nocturnal $O_3$ enhancement (NOE) on 31 July 2021 over the North China Plain is caused by the stratosphere $O_3$ intrusion after the passage of Typhoon In-fa. The authors argue that the NOE is instead due to the photochemically produced $O_3$, which can be brought downward from the residual layer at night. Overall, the manuscript is well-written, and the method is reasonable. I have a few points that I think could be addressed to strengthen the manuscript and some minor comments.

**Response: We thank the reviewer for the positive comments and suggestions. Here are our point-to-point responses to the comments**.

General Comments:

1. Although I tend to believe that STT usually has little effect on ground $O_3$ at low altitudes, the reasoning in the manuscript needs to be further elaborated. Could you explain more on why higher nighttime Ox than daytime $O_3$ is an indicator of downward mixing from the residual layer and vice versa? Also, both $O_3$ and PAN can be transported over long distances, and I am not sure why the in-phase changes of $O_3$ and PAN could exclude the possibility of STT. Lastly, the authors mention that HOLWCO could also occur in photochemically aged air, not necessarily from STT, but the photochemical age analysis in the manuscript shows fresh air with ages less than 10 hrs. Does that mean HOLWCO can occur in fresh air as well?

● Could you explain more on why higher nighttime Ox than daytime O3 is an indicator of downward mixing from the residual layer and vice versa?

**Response: Our intention is to check whether or not surface $O_3$ observed during the**

NOE contained significant contribution of $O_3$ from the stratosphere. For this purpose, a comparison of $O_3$ levels in the vertical direction is helpful. However, $O_3$ is very reactive and can be significantly removed by titration reactions in the boundary layer, with $O_3 + NO = NO_2 + O_2$ being the most important one. Therefore, $O_x$ ($O_3+NO_2$) is a more conserved quantity than $O_3$ and hence a better metric for comparison (Kley et al., 1994; Kleinmann et al., 2002; Caputi et al., 2019; He et al., 2022). During daytime, $NO_2$ formed in the titration reaction is rapidly photolyzed to regenerate $O_3$ so that the net chemical loss of $O_3$ is relatively small. At night, however, the reaction between $O_3$ and NO leads to lower levels of $O_3$ in the nocturnal boundary layer (NBL) and in the residual layer (RL). Because the emission of NO takes place mainly in the NBL, much more $O_3$ is removed by the titration reaction in the NBL than in the RL (Wang et al., 2018; Caputi et al., 2019; He et al., 2022). In addition, $O_3$ in the NBL is subjected to dry deposition. Therefore, the $O_3$ level before sunset largely remains in the RL (Caputi et al., 2019; He et al., 2022) and is usually much higher in the RL than in the NBL under normal conditions.

Following the method of He et al. (2022), we make comparison of afternoon $O_3$ averages on 31 July with the respective $O_x$ averages in the NBL during the NOE between 31 July and 1 August, 2021. To facilitate the comparison, we treat the average surface $O_3$ during 14:00-17:00 LT of 31 July as afternoon average of $O_3$ in the convective boundary layer, denoted as $[O_3]_{aft}$. Let us now focus on three nighttime atmospheric conditions, (I) undisturbed, (II) disturbed with NOE but no STT impact, and (III) disturbed with NOE and significant STT impact.

Under undisturbed condition (I), the nighttime average $O_3$ concentration in the RL ($[O_3]_{RL}$) should be close to (or only slightly lower than) $[O_3]_{aft}$ (Caputi et al., 2019; He et al., 2022), while the average $O_3$ concentration in the NBL ($[O_3]_{NBL}$) should be much lower than $[O_3]_{aft}$ due to the impacts of NO titration ($\Delta[O_3]_{titr}$) and dry deposition ($\Delta[O_3]_{dep}$), and the average $O_x$ concentration in the NBL ($[O_x]_{NBL}$) should also be lower than $[O_3]_{aft}$ due to dry deposition. The following relationships should be tenable:

$$[O_3]_{RL} \leq [O_3]_{aft} \qquad (1)$$
$$[O_3]_{NBL} = [O_3]_{aft} - \Delta[O_3]_{titr} - \Delta[O_3]_{dep} \qquad (2)$$
$$[O_x]_{NBL} = [O_3]_{aft} - \Delta[O_3]_{dep} \qquad (3)$$

Under disturbed condition with NOE but no STT impact (II), a downward transport of $O_3$ from the RL to NBL should be considered. Assuming that the downward transport causes a reduction of $[O_3]_{RL}$ by $\Delta[O_3]_{D1}$ and an increase of $[O_3]_{NBL}$ by $\Delta[O_3]_{D2}$, then

$$[O_3]_{RL} \leq [O_3]_{aft} - \Delta[O_3]_{D1} \qquad (4)$$

$$[O_3]_{NBL} = [O_3]_{aft} - \Delta[O_3]_{titr} - \Delta[O_3]_{dep} + \Delta[O_3]_{D2} \tag{5}$$

$$[O_x]_{NBL} = [O_3]_{aft} - \Delta[O_3]_{dep} + \Delta[O_3]_{D2} \tag{6}$$

Under disturbed condition with NOE and STT impact (III), net contributions of $O_3$ from the STT should be considered to the RL and the NBL. Assuming that the STT contribution increases $[O_3]_{RL}$ and $[O_3]_{NBL}$ by $\Delta[O_3]_{STT1}$ and $\Delta[O_3]_{STT2}$, respectively, then

$$[O_3]_{RL} \leq [O_3]_{aft} + \Delta[O_3]_{STT1} \tag{7}$$

$$[O_3]_{NBL} = [O_3]_{aft} - \Delta[O_3]_{titr} - \Delta[O_3]_{dep} + \Delta[O_3]_{STT2} \tag{8}$$

$$[O_x]_{NBL} = [O_3]_{aft} - \Delta[O_3]_{dep} + \Delta[O_3]_{STT2} \tag{9}$$

Equation (3) indicates that $[O_x]_{NBL}$ should be significantly lower than $[O_3]_{aft}$ under undisturbed conditions. Although equation (6) shows that $[O_x]_{NBL}$ could be higher than $[O_3]_{aft}$ (i.e., if $\Delta[O_3]_{dep} < \Delta[O_3]_{D2}$), it cannot really occur because $[O_3]_{RL} < [O_3]_{aft}$ (see equation (4)) and $O_3$ cannot be transported from a lower concentration position to the higher one. Therefore, $[O_x]_{NBL}$ should not be significantly higher than $[O_3]_{aft}$ under disturbed conditions with NOE but no STT impact. Dry deposition is only a small sink for nighttime surface $O_3$ in northern China (Tang et al., 2017), while a STT impact could substantially enhanced the level of surface $O_3$ if it reaches the surface layer. Hence, it is very likely according to equation (9) that $[O_x]_{NBL}$ is significantly higher than $[O_3]_{aft}$ under disturbed condition with a STT impact. In summary, $[O_x]_{NBL}$ should be significantly higher than $[O_3]_{aft}$ if the NBL is really impacted by stratospheric $O_3$, otherwise the STT impact is negligible even though a NOE event is observed.

As shown in Table 1 in our manuscript, in all cities but Weihai (WH), mean $O_x$ during the maximum NOE period was either much lower than or nearly equal to mean $O_3$ during the afternoon period. This indicates that $O_3$ in the NBL in those cities was probably not impacted by stratospheric $O_3$. Mean $O_x$ in WH during the NOE was much higher than mean $O_3$ in the erstwhile afternoon. However,this could not be attributed to stratospheric impact. Chen et al. (2022) mentioned "the surface ozone enhancement at midnight was not coincident with CO reduction, suggesting that the stratospheric air mass might not have reached the surface." (at the end of the second paragraph in section 4.2). The higher $[O_x]_{NBL}$ in WH may also be interpreted by the above equation (6), i.e., if $\Delta[O_3]_{D2}$ was much greater than $\Delta[O_3]_{dep}$. A great $\Delta[O_3]_{D2}$ would require a high $[O_3]_{aft}$, either from local photochemical production or from transport. Actually, photochemically produced $O_3$ (PPO) had not been well established in WH during the daytime on 31 July, as can be seen in our Figure 1f. Therefore, regional-scale transport of PPO is suspected. In section 3 of our manuscript, we provided an interpretation in the viewpoint of diurnal alternations of land-sea breezes. Our view is supported by the backward trajectories shown in Figure 1R(right), which indicate that air parcels influenced the

NBL in WH were mostly from the Yellow Sea area. The prevailing wind direction at 850 hPa over the Yellow Sea and the neighboring land was SW, as shown in Figure 5 (a) in Chen et al. (2022). Such wind condition could facilitate the transport of PPO from the continent to the sea area. Because of the lower emissions of $NO_x$ over the sea, PPO can be well sustained at night and transported to continental locations like WH through sea breezes.

In Figure R1, we also show 24-h backward trajectories arriving at 100 m above ground over ZB and BZ. All the trajectories do not indicate any transport of air parcels from altitudes over daytime boundary layer. Figure R2 shows a matrix of backward trajectories for air parcels arriving at 100 m above ground over the domain 36°-38°N and 115°-122°E at 19:00 UTC, 31 July 2021. Only 3 of the 24 trajectories were ever travelled over daytime boundary layer and the 3 trajectories ended at locations over the Bohai Gulf.

[Figure]

**Figure R1. Backward trajectories for air parcels arriving at 100 m above ground over Zibo (ZB, left), Binzhou (BZ, middle) and Weihai (WH, right) every hour between 19:00 and 08:00 UTC, 31 July 2021. The trajectories were computed online using the HYSPLIT model (https://www.ready.noaa.gov/HYSPLIT.php; Stein et al. 2015; Rolph et al., 2017) and the Global Forecast System (GFS) reanalysis data (0.25° resolution, https://www. emc.ncep. noaa.gov/emc/pages/numerical_forecast_systems/gfs.php). The total run time for the trajectories was 24 hours.**

[Figure]

**Figure R2. Matrix of backward trajectories for air parcels arriving at 100 m above ground over the domain 36°-38°N and 115°-122°E at 19:00 UTC, 31 July 2021. The trajectories were computed online using the HYSPLIT model (https://www.ready.noaa.gov/HYSPLIT.php; Stein et al. 2015; Rolph et al., 2017) and the Global Forecast System (GFS) reanalysis data (0.25° resolution, https://www.emc.ncep.noaa.gov/emc/pages/numerical_forecast_systems/gfs.php). The total run time for the trajectories was 24 hours.**

**In addition to the backward trajectories, some ground-based remote sensing data also support our view. Figure R3 presents time-altitude cross sections of $O_3$, relative humidity, virtual temperature and wind speed observed at the ZB supersite between 18:00 LT, 31 July and 06:00 LT, 1 August 2021. As can been seen in the figure, relatively higher $O_3$ mixing ratios occurred only below about 0.5 km. From the evening to midnight, very strong wind prevailed below 1 km, after which humidity was largely enhanced, it rained and the NOE event occurred. The top of the higher $O_3$ layer was partly uplifted during the NOE period. According to the backward trajectories shown in Figure R2 (left), the air parcels were from the southwest sector and travelled mostly below 0.5 km above sea level. There was no indication of air from the free troposphere. And it is very likely that the NOE was due to horizontal transport of daytime PPO from the southwest sector.**

[Figure]

**Figure R3. Time-altitude cross sections of O₃, relative humidity (RH), virtual temperature (Tv) and wind speed (WS) observed at the ZB supersite between 18:00 LT, 31 July and 06:00 LT, 1 August 2021.**

In conclusion, our comparisons of nighttime $O_x$ with daytime $O_3$ in the NCP cities and analysis of backward trajectories and remote sensing data suggest that the NOE events observed at the NCP cities were most likely from the transport of $O_3$ in the RL or (in the case of WH) in the marine boundary layer, and unlikely from the transport of stratospheric $O_3$.

We included the discussion about the comparison between afternoon $O_3$ and $O_x$ in the NBL in the revised supplement as Text S1 (lines 77-122), changed the original Text S1 to Text S2 and renumbered the equations and figures.

**"Text S1 Identifying potential origin of the NOE by comparing afternoon $O_3$ with $O_x$ in the NBL during the NOE**

Our intention is to check whether or not surface $O_3$ observed during the NOE contained significant contribution of $O_3$ from the stratosphere. For this purpose, a comparison of $O_3$ levels in the

vertical direction is helpful. …….In summary, $[O_x]_{NBL}$ should be significantly higher than $[O_3]_{aft}$ if the NBL is really impacted by stratospheric $O_3$, otherwise the STT impact is negligible even though a NOE event is observed."

**In section 3.2 in the revised manuscript, we cited Text S1 and changed text as follows:**

"Following the method of He et al. (2022), we make comparison of $O_3$ averages during 14:00-17:00 LT on 31 July in the above cities with the respective $O_x$ ($O_3+NO_2$) averages during the periods of the maximum NOE between 31 July and 1 August (Table 1). Such comparison facilitates the judgment whether or not the NOE was caused by downward mixing of air in the residual layer (RL) into the nocturnal boundary layer (NBL) because afternoon averages of $O_3$ in the convective boundary layer are well preserved at night in the RL and $O_x$ is a more conserved quantity than $O_3$ in the NBL. Details about the reasonability of this method are given in Text S1. It can be seen in Table 1 that, except for WH, the nighttime $O_x$ averages approach to or obviously lower than the respective daytime $O_3$ averages. In the mega-cities QD and JN, the average levels of $O_x$ during the maximum NOE were nearly the same as those of daytime $O_3$, while the nighttime $O_x$ in the other cities (excluding WH) was at least a few ppbv lower than daytime $O_3$. According to the discussions in Text S1, for all cities excluding WH, data in Table 1 do not suggest any significant STT impact on the NOE events."

**We included Figures R1 and R2 in the revised supplement as Figures S3 and S4. We cited Figures S3 and S4 in section 3.3 in the revised manuscript and changed the text about influences of land-sea breezes on WH:**

"The case of WH deserves more detailed analysis. WH is a relatively smaller city in the tip of the Shandong Peninsula. The higher $O_x$ concentration for WH in Table 1 was probably related to the regional transport of air pollution and the influence of the diurnal alternations of land-sea breezes. Actually, the afternoon PPO in WH had not been well established as shown in Figure 1f. During the daytime (particularly afternoon), when sea breeze dominates, PPO is significantly diluted by cleaner air from the marine boundary layer. The daily maximum of $O_3$ in WH is generally observed between 11:00 LT and 13:00 LT, rather than between 14:00 LT and 17:00 LT. If the $O_3$ average for WH in Table 1 were replaced with that from 11:00 LT to 13:00 LT on 1 August (80 ± 2.0 ppbv), then the difference between $O_x$ and $O_3$ would be reduced to about 21 ppbv. At night, when the land breeze dominates, the near surface level of WH is usually controlled by divergence, which induces downdraft from the residual layer, transports daytime PPO residing in the residual layer to the surface, and resulted in the NOE. This might have been the main reason of highly frequent NOE emerging in WH. "

**We added two paragraphs in section 3.3 to extend our discussion on atmospheric transport:**

"Chen et al. (2022) investigated the atmospheric transport process of the NOE by using high-resolution Weather Research and Forecasting (WRF) simulation and FLEXible PARTicle (FLEXPART) particle dispersion modelling. They presented the two scenarios for BZ and QD and found very different results for the two cities. To support above view, we calculated backward

trajectories of air parcels arriving at 100 m above ground level over WH every hour between 19:00 and 08:00 UTC, 31 July 2021 using HYSPLIT model (https://www.ready.noaa.gov/HYSPLIT.php) and the Global Forecast System (GFS) reanalysis data (0.25 °resolution, https://www.emc.ncep.noaa.gov/emc/pages/numerical_forecast_systems/gfs.php). Our intention is not to resolve the dynamical evolution of the MCSs but to analyze atmospheric transport at a relatively larger scale. Figure S3 shows the calculated backward trajectories for WH, together with those for ZB and BZ in the same time window. The trajectories in Figure S3(right) indicate that air parcels influencing the NBL in WH were mostly from the marine boundary layer over the Yellow Sea area. The prevailing wind direction at 850 hPa over the Yellow Sea and the neighbouring land was SW, as shown in Figure 5a in Chen et al. (2022). Such wind condition could facilitate the transport of PPO from the continent to the sea area. Because of the lower emissions of $NO_x$ over the sea, PPO can be well sustained at night and transported to continental locations like WH through sea breezes.

The 24-h backward trajectories for ZB (Figure S3(left)) and BZ (Figure S3(middle)) provide additional clues denying STT impacts on nighttime $O_3$ in these cities. All the trajectories do not indicate any transport of air parcels from altitudes over daytime boundary layer. To gain a more complete insight into the air movements during and before the NOE events, we show in Figure S4 a matrix of backward trajectories for air parcels arriving at 100 m above ground level over the domain 36 °38 °N and 115 °122 °E at 19:00 UTC (03:00 LT), 31 July 2021. The trajectory heights and locations shown in Figure S4 indicate that only 3 of the 24 trajectories travelled over daytime boundary layer and the 3 trajectories ended at locations over the Bohai Gulf. Therefore, our systematic trajectory analysis does not suggest that the NOE events in the NCP cities were related with downward transport of airmasses from the free troposphere."

**We included Figure R3 in the revised supplement as Figure S8 and added profiles of wind vector as Figure S9. We made section 3.6 in the revised manuscript to cite these figures and discuss the results as follows:**

"In addition to the backward trajectories, some ground-based remote sensing data from ZB also support our view. Figure S8 presents time-altitude cross sections of $O_3$, relative humidity (RH), virtual temperature (Tv) and wind speed (WS) observed at the ZB supersite between 18:00 LT, 31 July and 06:00 LT, 1 August 2021. As can been seen in the figure, relatively higher $O_3$ mixing ratios occurred only below about 0.5 km. From the evening to midnight, very strong wind prevailed below 1 km, after which humidity was largely enhanced, it rained and the NOE event occurred. The wind directions below 1 km were southerly before 01:00 LT and turned to northerly by 02:00 LT (Figure S9). The top of the higher $O_3$ layer was partly uplifted during the NOE period. According to the backward trajectories shown in Figure S3(left), the air parcels were from the southwest sector and travelled mostly below 0.5 km above sea level. Both the remote sensing data and the trajectory analysis provide no evidence of air from the free troposphere. Therefore, it is very likely that the NOE observed from the ground to about 0.5 km was due to advection transport of daytime PPO from the southwest sector rather than SST impacts."

- Also, both $O_3$ and PAN can be transported over long distances, and I am not sure why

the in-phase changes of $O_3$ and PAN could exclude the possibility of STT.

**Response: Indeed, both $O_3$ and PAN can be transported over long distances at higher altitudes, where their chemical removal is limited. However, PAN is primarily produced where VOCs and $NO_x$ are largely emitted, such as in industrial and populated areas or in biomass burning plumes (Xu et al., 2018). Higher temperatures and intense radiation are favorable meteorological conditions for the formation of PAN. These chemical and meteorological conditions facilitate the production of $O_3$ as well. For this reason, $O_3$ and PAN in the boundary layer are usually well correlated, particularly in warm seasons. The in-phase changes of $O_3$ and PAN reflect the dominant influences of chemical and physical processes within the boundary layer but do not exclude the possibility of STT. What really do not support the possibility of STT is the relatively higher value of PAN observed during the NOE at the ZB supersite. The average PAN (0.44±0.02 ppbv) was higher than the lowest nighttime PAN levels on most days before and after the NOE event. This evidence disproves the possibility of significant STT impact.**

**Lacking of parallel measurements of PAN at high altitudes over the NCP hinders us from making a vertical comparison of PAN levels. However, some PAN values from earlier high altitude observations can be referenced. The PAN values in the UTLS over the NCP in August 2003 ranged from about 0.17 ppb (at 185 hPa) to 0.35 ppb (at 278 hPa), as retrieved from the satellite observations (Moore and Remedios, 2010). Ten-years of satellite observations showed that average PAN in the UTLS over the anticyclone region (60–120 E, 10–40 N) reached its maximum in summer and varied in the range of 0.06-0.3 ppb (Fadnavis et al., 2014). Aircraft measurements of PAN in July 2008 over Longyearbyen, Spitsbergen (78 °130′N, 15 °330′E) and Oberpfaffenhofen, Germany (48 °40′N, 11 °160′E) showed PAN levels in the range of 0.05-0.3 ppb in the free troposphere and lower stratosphere (Roiger et al., 2011). Airborne measurements over North America showed summer PAN in the free troposphere and lowermost stratosphere were usually lower than 0.5 ppb unless biomass burning air masses were sampled (Singh et al., 2007). Observations at high mountain sites also indicated quite low PAN levels in the free troposphere. For example, multiyear observations at the high alpine site Jungfraujoch (46.55 N, 7.98 E, 3580 ma.s.l) found average summer PAN levels of lower than 0.2 ppb in free tropospheric air masses (Pandey Deolal et al., 2013) and those at Summit station, Greenland (72.34 N, 38.29 W, 3212 ma.s.l.) showed average summer PAN levels of lower than 0.1 ppb (Kramer et al., 2015). In summary, the mixing ratios of PAN in the free troposphere and lower stratosphere of the Northern Hemisphere are normally much lower than 0.5 ppb and mostly lower than 0.3 ppb.**

**PAN is thermally unstable and decomposes rapidly in warm urban air. According to Cox and Roffey (1977), the lifetime of PAN was only 2.7 hours at 25 °C. In our case,**

the nighttime temperature at the ZB supersite varied from 33.1 °C at 18:00 LT on 31 July to 20.4 °C at 06:00 LT on 1 August, 2021, with an average of 26.0 °C. Under such warm condition, even shorter lifetimes of PAN are expected. The thermal decomposition lifetimes were in the range of 0.2-1.3 hours based on the hourly observations. If the surface layer had been significantly impacted by air masses from the free troposphere and lower stratosphere, we would have seen much lower levels of PAN during the NOE instead of the observed 0.4-0.5 ppb (Figure 2). Therefore, given the relative lower concentrations of PAN in the free troposphere and lower stratosphere and the rapid thermal decomposition, it is unlikely that over 0.4 ppb of PAN could be observed in surface air significantly impacted by stratosphere intrusion, not to mention that the PAN values during the NOE were even much higher than the nighttime values of other dates.

**To include our major points on this issue, we extended the discussion in section 3.7 in the revised manuscript on the PAN and O$_3$ measurement from the ZB supersite:**

"To further understand the source characteristic of NOE, the hourly average concentrations of surface PAN and O$_3$ observed at the ZB supersite are displayed in Figure 2. It can be seen that the variations of PAN and O$_3$ were in phase and the concentrations of both gases were well correlated ($r^2$=0.60, p<0.0001, n=283), indicating that the variations of PAN and O$_3$ were driven mainly by chemical and physical processes within the boundary layer. The maximum O$_3$ from 02:00 LT to 05:00 LT on 1 August was 88 ± 5 ppbv, matching with PAN of 0.44 ± 0.02 ppbv, which was still significantly higher than those lowest values in nighttime of other dates. As a secondary photochemical pollutant, PAN is thermally unstable but can be transported over higher altitudes, where it has much longer lifetime due to low temperatures. However, observations over the years on mountain tops, aircrafts and satellites showed that the mixing ratios of PAN in the free troposphere and lower stratosphere of the Northern Hemisphere were normally much lower than 0.5 ppb and mostly lower than 0.3 ppb (Singh et al., 2007; Moore and Remedios, 2010; Roiger et al., 2011; Pandey Deolal et al., 2013; Fadnavis et al., 2014; Kramer et al., 2015). PAN decomposes rapidly in warm urban air. According to Cox and Roffey (1977), the lifetime of PAN was only 2.7 hours at 25 °C. In our case, the nighttime temperature at the ZB supersite varied from 33.1 °C at 18:00 LT on 31 July to 20.4 °C at 06:00 LT on 1 August, 2021, with an average of 26.0 °C. Under such warm condition, the thermal decomposition lifetimes were in the range of 0.2-1.3 hours based on the hourly observations. If the surface layer had been significantly impacted by air masses from the free troposphere and lower stratosphere, we would have seen much lower levels of PAN during the NOE instead of the observed 0.4-0.5 ppb (Figure 2). Therefore, given the relative lower concentrations of PAN in the free troposphere and lower stratosphere and the rapid thermal decomposition, it is unlikely that over 0.4 ppb of PAN could be observed in surface air significantly impacted by stratospheric intrusion, not to mention that the PAN values during the NOE were even much higher than the nighttime values before and after the NOE event."

- Lastly, the authors mention that HOLWCO could also occur in photochemically aged air, not necessarily from STT, but the photochemical age analysis in the manuscript

shows fresh air with ages less than 10 hrs. Does that mean HOLWCO can occur in fresh air as well?

**Response: Chen et al. (2022) attributed the ozone-rich and CO-poor phenomenon during the NOE to the downward transport of stratospheric air. In our manuscript we mentioned "photochemically aged pollution air masses may also show HOLWCO features as observed in the PEM-West A campaign (Newell et al. 1996b; Stoller et al., 1999)" and "air mass with HOLWCO feature in free troposphere do not necessarily mean that the $O_3$ enhancement is originated from the stratosphere (Stoller et al., 1999)". Since HOLWCO phenomena may not be caused by STT, we suggest that more cautions should be taken when attributing HOLWCO events at low altitude sites to stratospheric impact.**

**The reviewer is asking the possibility of HOLWCO occurring in fresh air, referring the relatively shorter lifetimes we estimated. Our results can neither confirm nor exclude this possibility. Our estimates of photochemical age indicate that surface air at ZB during the NOE was nearly as fresh as at other nights, against a significant impact of stratospheric air, which has ages of years. In addition, the HOLWCO feature during the NOE was very much far from the stratospheric characteristics. The maximum $O_3$ levels were around 80 ppb and significantly lower than respective daytime maxima; the CO levels were between 200 and 500 ppb (see Figure 3 in Chen et al., 2022 and Figure R4), much higher than the CO levels in the middle and upper troposphere (about 100 ppb) and lower stratosphere (<50 ppb) (Inness et al., 2022) and even not lower than those CO values on some other days; the measured water vapor pressure during the NOE was close to its normal values (Figure R4) and did not show any sign of substantial stratospheric impact. In other words, although the levels of CO or water vapor were relatively lower during the NOE event, they did not show deviations from their values within normal boundary layer, nor substantial STT influences. All these, together with our evidence given above, indicate that the NOE reported in Chen et al. (2022) was not caused by typhoon-induced stratospheric intrusion.**

[Figure]

**Figure R4. Time series of hourly multi-sites-averages of O₃ (purple), CO (black ) and water vapor pressure (blue) between 25 July and 5 August 2021. Data from 18:00 LT on 31 July to 06:00 LT on 1 August are highlighted in thick lines. The multi-site data from HS (4 sites) (a), BZ (6 sites) (b), JN (13 sites)(c), WF (6 sites) (d), QD (13 sites) (e), WH (5 sites)(f) and ZB (7sites) (g) are available at https://quotsoft.net/air (last access: 15 April 2023; X. L. Wang, 2020). Data from the ZB supersite (h) is provided by the Chinese Academy of Environmental Sciences (CRAES). Water vapor pressure data are from https://data.cma.cn/.**

[Figure]

**Figure R4. (continued).**

**We included Figure R4 in the revised supplement as Figure S6 and added some text in section 4 in the revised manuscript to cite and discuss the data in this figure:**

"In the NOE cases reported in Chen et al. (2022), the HOLWCO feature during the NOE was very much far from the stratospheric characteristics. The maximum $O_3$ levels were around 80 ppb and significantly lower than respective daytime maxima; the CO levels were between 200 and 500 ppb (see Figure 3 in Chen et al., 2022 and Figure S6), much higher than the CO levels in the middle and upper troposphere (about 100 ppb) and lower stratosphere (<50 ppb) (Inness et al., 2022) and even not lower than those CO values on some other days; the measured water vapor pressure during the NOE was close to its normal values (Figure S6) and did not show any sign of substantial stratospheric impact. In other words, although the levels of CO or water vapor were relatively lower during the NOE events, they did not show large deviations from their values within normal boundary layer, nor substantial STT influences. All these, together with our evidence given above, indicate that the NOE reported in Chen et al. (2022) was not caused by typhoon-induced stratospheric intrusion. Therefore, more cautions should be taken when attributing HOLWCO events at low altitude sites to stratospheric impact, whether or not there was an influence from a typhoon or other synoptic system. It is suggested that if possible, each case should be verified by analyzing both physical and chemical processes before making a

conclusion."

2. If possible, the vertical profiles of virtual temperature and winds can be added to further support the conclusion that the rapid downward transport of daytime PPO.

**Response. There are three routine radiosonde stations in the studied area, i.e., Zhangqiu (ZQ, 117.524 °E, 36.713 °N), Rongcheng (RC, 122.477 °E, 37.173 °N) and QD. ZQ is about 52 km east of JN and RC about 56 km southeast of WH. The radiosonde data collected at 19:00 LT, 31 July and 07:00 LT, 1 August 2021 at these sites can be used to get a glimpse of the vertically thermal and dynamical evolutions in the night of 31 July. The raw radiosonde data include temperature, pressure, relative humidity, and wind speed and direction (https://data.cma.cn/). We calculated the virtual temperature and equivalent potential temperature($\theta_{se}$) (Bolton, 1980) and wind shear, $(du/dz)^2$ (Cho et al., 2001). These vertical profiles of these quantities from surface level to 400 hPa are shown in Figure R5. The pronounced decreasing of $\theta_{se}$ below 900 hPa over ZQ and QD from 19:00 LT, 31 July to 07:00 LT, 1 August indicated a descending process occurred at the night. The wind shear peaked near 900 hPa over ZQ and QD, providing kinetic energy for the mixing process. The above thermal and dynamical conditions were, of course, favorable for the downward mixing of higher levels of $O_3$ in the residual layer over ZQ and QD. Over RC, however, the thermal and dynamical conditions were different (Figure R5c) and less favorable for triggering the downward transport. This is consistent with the data from the neighboring coast city WH, showing high surface $O_3$ during the NOE event accompanied with relatively high CO and water vapor (Figure R4f).**

[Figure]

**Figure R5.** The vertical profiles of virtual temperature (red), equivalent potential temperature (θse, grey) and wind shear (blue) calculated on the basis of routine radiosonde observations data at 19:00 LT, 31 July and 07:00 LT, 1 August 2021 in Zhangqiu (ZQ)(a), Qingdao (QD )(b) and Rongcheng (RC) (c).

**We included Figure R5 in the revised supplement as Figure S5 and added section 3.4 in the revised manuscript to discuss the data in this figure:**

"To confirm the possibility of rapid downward transport of daytime PPO, we obtained some radiosonde data from three stations in Shandong Province, i.e., Zhangqiu (ZQ, 117.524 °E, 36.713 °N), Rongcheng (RC, 122.477 °E, 37.173 °N) and QD. ZQ is about 52 km east of JN and RC about 56 km southeast of WH. The radiosonde data collected at 19:00 LT, 31 July and 07:00 LT, 1 August 2021 at these sites can be used to get a glimpse of the vertically thermal and dynamical

evolutions in the night of 31 July. The raw radiosonde data include temperature, pressure, relative humidity, and wind speed and direction (https://data.cma.cn/). We calculated the virtual temperature and equivalent potential temperature ($\theta se$) (Bolton, 1980) and wind shear, $(du/dz)^2$ (Cho et al., 2001). The vertical profiles of these quantities from surface level to 400 hPa are shown in Figure S5. The pronounced decreasing of $\theta se$ below 900 hPa over ZQ and QD from 19:00 LT, 31 July to 07:00 LT, 1 August indicated that a descending process occurred at the night. The wind shear peaked near 900 hPa over ZQ and QD, providing kinetic energy for the mixing process. The above thermal and dynamical conditions were favorable for the downward mixing of higher levels of $O_3$ in the residual layer over ZQ and QD. Over RC, however, the thermal and dynamical conditions were different (Figure S5c) and less favorable for triggering the downward transport. This is consistent with the data from the neighboring coast city WH, showing high surface $O_3$ during the NOE event accompanied with relatively high CO and water vapor (Figure S6f). As a coast city near WH, RC should have been impacted by airmasses from marine boundary layer, as discussed for WH in section 3.3."

Minor Comments:

Line 43: 'rangeby' should be 'range by'.

**Response: Thank you. This error was corrected.**

Line 66-67: Better to add how many sites there are in each city to the main texts and mention that Fig. 1 is the site average.

**Response: Yes, the site numbers were added in the revised manuscript.**

"Figure 1: Time series of hourly multi sites-average of $O_3$ (purple) and $NO_2$ (bright blue) in several NCP cities between 25 July and 5 August 2021. Data from 18:00 LT on 31 July to 06:00 LT on 1 August are highlighted in red. The multisite data from HS (4 sites) (a), BZ (6 sites) (b), JN (13 sites) (c), WF (6 sites) (d), QD (13 sites) (e), WH (5 sites) (f) and ZB (7 sites) (g) are available at https://quotsoft.net/air (last access: 15 April 2023; X. L. Wang, 2020). Data from the ZB supersite (h) is provided by the Chinese Academy of Environmental Sciences (CRAES). The positive (negative) error bars represent one standard deviation of $O_3$($NO_2$)."

Line 87-90: Consider putting an enlarged figure in the supplement and mark these NOEs. It is hard to identify in Figure 1.

**Response : Yes, Figure S2 was included in the revised supplement to show an enlarged view of the NOEs.**

Line 127: Change 'arrived' to 'arriving'.

The caption of Figure 3: Change 'arrived' to 'arriving' and 'based' to 'based on'.

**Response : Corrected (line 221).**

whole process of the convective activity because it only show six snapshots of the mesoscale convective system (MSC) observed between 20:00 LT, 31 July and 01:00 LT, 1 August 2021 and does not show the dissipation of the MSC. The radar reflectivity maps in Figure R6 add additional snapshots to the MSC. As can be seen in Figures R6b-R6f, Zibo and Jinan were clearly under the influence of the MSC around 00:00 LT, hited heavily by the MSC between 01:00 LT and 03:00 LT, 1 August. Without doubt, the MSC impacting the cities with NOEs reported in Chen et al. (2022) also impacted Zibo. The radar reflectivity maps show clearly that except QD and WH, all cities listed in our Table 1 were strongly impacted by the MSC. While the occurrence of the NOE was a little later than the MSC impact, the sequence of the MSC impacts is consistent with that of the NOE events. Therefore, the major conclusions based on our analysis apply also to Chen et al. (2022).

[Figure]

Figure R6. Radar reflectivity maps for the North China Plain at (a) 15:06 UTC (23:00 LT, 31 July), (b) 16:12 UTC (00:12 LT, 1 August), (c) 17:00 UTC (01:00 LT, 1 August), (d) 18:06 UTC (02:06 LT, 1 August),

**We included Figure R6 in the revised supplement as Figure S7 and added section 3.5 in the revised manuscript to discuss this issue:**

"Above analyses show that not all cities in the study area were clearly influenced by strong downward transport. The cities strongly impacted by the MCSs should have experienced intensive vertical air motion. Chen et al. (2022) showed in their Figure 8 that the vertical atmospheric activity seemed to be limited near ZB and the main convective zone was in the western Shandong Province. However, this figure presents by far not the whole process of the convective activity because it only show six snapshots of the MSCs observed between 20:00 LT, 31 July and 01:00 LT, 1 August 2021 and does not show the dissipation of the MSCs. The radar reflectivity maps in Figure S7 add additional snapshots to the MSCs. As can be seen in Figure S7b-f, ZB and JN were clearly under the influence of the MSCs around 00:00 LT, hit heavily by the MSCs between 01:00 LT and 03:00 LT, 1 August. Without doubt, the MSCs impacting the cities with NOEs reported in Chen et al. (2022) also impacted ZB. The radar reflectivity maps show clearly that except QD and WH, all cities listed in our Table 1 were strongly impacted by the MSCs. While the occurrence of the NOE was a little later than the MSCs impact, the sequence of the MSCs impacts is consistent with that of the NOE events. Therefore, the major conclusions based on our analysis of data from the ZB supersite should also apply to Chen et al. (2022). In next sections, we offer more evidence from ZB to support our argument."

One small comment is to clarify $O_3$ as "daytime $O_3$" and Ox as "nighttime Ox" in Table 1 **Response: Thank you. We added clarifications in the revised manuscript and supplement.**

**We included the discussion about the comparison between afternoon $O_3$ and $O_x$ in the NBL in the revised supplement as Text S1 (lines 77-122), changed the original Text S1 to Text S2 and renumbered the equations and figures.**

"**Text S1 Identifying potential origin of the NOE by comparing afternoon $O_3$ with $O_x$ in the NBL during the NOE**

Our intention is to check whether or not surface $O_3$ observed during the NOE contained significant contribution of $O_3$ from the stratosphere. For this purpose, a comparison of $O_3$ levels in the vertical direction is helpful. …….In summary, $[O_x]_{NBL}$ should be significantly higher than $[O_3]_{aft}$ if the NBL is really impacted by stratospheric $O_3$, otherwise the STT impact is negligible even though a NOE event is observed."

**In section 3.2 in the revised manscript, we cited Text S1 and changed text as follows:**

"Following the method of He et al. (2022), we make comparison of $O_3$ averages during 14:00-17:00 LT on 31 July in the above cities with the respective $O_x$ ($O_3+NO_2$) averages during the periods of the maximum NOE between 31 July and 1 August (Table 1). Such comparison facilitates the judgement whether or not the NOE was caused by downward mixing of air in the

residual layer (RL) into the nocturnal boundary layer (NBL) because afternoon averages of $O_3$ in the convective boundary layer are well preserved at night in the RL and $O_x$ is a more conserved quantity than $O_3$ in the NBL. Details about the reasonability of this method are given in Text S1. It can be seen in Table 1 that, except for WH, the nighttime $O_x$ averages approach to or obviously lower than the respective daytime $O_3$ averages. In the mega-cities QD and JN, the average levels of $O_x$ during the maximum NOE were nearly the same as those of daytime $O_3$, while the nightime $O_x$ in the other cities (excluding WH) was at least a few ppbv lower than daytime $O_3$. According to the discussions in Text S1, for all cities excluding WH, data in Table 1 do not suggest any significant STT impact on the NOE events."

**In addition to above changes, we made minor changes in abstract, added references cited in the reference lists, modified Figure 1 by showing only one side error-bars and corrected any errors we found. Please see more details in our marked-up manuscript and supplement.**

[revised manuscript text omitted]

Figure S3. Backward trajectories for air parcels arriving at 100 m above ground over Zibo (ZB, left), Binzhou (BZ, middle) and Weihai (WH, right) every hour between 19:00 and 08:00 UTC, 31 July 2021. The trajectories were computed online using the HYSPLIT model (https://www.ready.noaa.gov/HYSPLIT.php; Stein et al. 2015; Rolph et al., 2017) and the Global Forecast System (GFS) reanalysis data (0.25 ° resolution, https://www.emc.ncep.noaa.gov/emc/pages/numerical_forecast_systems/gfs.php). The total run time for the trajectories was 24 hours.

[Figure]

Figure S4. Matrix of backward trajectories for air parcels arriving at 100 m above ground over the domain 36 °-38 °N and 115 °-122 °E at 19:00 UTC, 31 July 2021. The trajectories were computed online using the HYSPLIT model (https://www.ready.noaa.gov/HYSPLIT.php; Stein et al. 2015; Rolph et al., 2017) and the Global Forecast System (GFS) reanalysis data (0.25 ° resolution, https://www.emc.ncep.noaa.gov/emc/pages/numerical_forecast_systems/gfs.php). The total run time for the trajectories was 24 hours.

[Figure]

**Figure S5. The vertical profiles of virtual temperature (red), equivalent potential temperature (θse, grey) and wind shear (blue) calculated on the basis of routine radiosonde observations data at 19:00 LT, 31 July and 07:00 LT, 1 August 2021 in Zhangqiu (ZQ)(a), Qingdao (QD )(b) and Rongcheng (RC) (c).**

[Figure]

**Figure S6. Time series of hourly multi-sites-averages of O$_3$ (purple), CO (black ) and water vapor pressure (blue) between 25 July and 5 August 2021. Data from 18:00 LT on 31 July to 06:00 LT on 1 August are highlighted in thick lines. The multi-site data from HS (4 sites) (a), BZ (6 sites) (b), JN (13 sites)(c), WF (6 sites) (d), QD (13 sites) (e),WH (5 sites)(f) and ZB ( 7sites) (g) are available at https://quotsoft.net/air (last access: 15 April 2023; X. L. Wang, 2020). Data from the ZB supersite (h) is provided by the Chinese Academy of Environmental Sciences (CRAES). Water vapor pressure data are from https://data.cma.cn/.**

[Figure]

**Figure S6. (continued).**

[Figure]

**Figure S7. Radar reflectivity maps for the North China Plain at (a) 15:06 UTC (23:00 LT, 31 July), (b) 16:12 UTC (00:12 LT, 1 August), (c) 17:00 UTC (01:00 LT, 1 August), (d) 18:06 UTC (02:06 LT, 1 August), (e) 19:00 UTC (03:00 LT, 1 August) and (f ) 20:00 UTC (04:00 LT, 1 August). Some of the major cities are indicated on the maps. The white star on each map shows the location of Zibo. The radar reflectivity maps were provided by Institute of Artificial Intelligence for Meteorology (IAIM), Chinese Academy of Meteorological Sciences (CAMS).**

[Figure]

**Figure S8. Time-altitude cross sections of O₃, relative humidity (RH), virtual temperature (Tv) and wind speed (WS) observed at the ZB supersite between 18:00 LT, 31 July and 06:00 LT, 1 August 2021. Vertical profiles of O₃ were obtained by a Lidar (RayOL-GB, Anhui Kechuang Zhongguang Technology Co., Ltd., China). Profiles of RH and Tv were detected by a microwave radiometer (KT-001, Qingdao Tianlang Environmental Technology Co., Ltd., China). Vertical distributions of WS were observed by a wind profiler radar (WindPrint V2000, Qingdao Huanhang Safety Environment Technology Co., Ltd, China).**

70

[Figure]

**Figure S9. Hourly averaged wind vector below 1 km over the ZB supersite between 18:00 LT, 31 July and 06:00 LT, 1 August 2021. Wind data were observed by a wind profiler radar (WindPrint V2000, Qingdao Huanhang Safety Environment Technology Co., Ltd, China).**

75

**Text S1 Identifying potential origin of the NOE by comparing afternoon $O_3$ with $O_x$ in the NBL during the NOE**

Our intention is to check whether or not surface $O_3$ observed during the NOE contained significant contribution of $O_3$ from the stratosphere. For this purpose, a comparison of $O_3$ levels in the vertical direction is helpful. However, $O_3$ is very reactive and can be significantly removed by titration reactions in the boundary layer, with $O_3 + NO = NO_2 + O_2$ being the most important one. Therefore, $O_x$ ($O_3+NO_2$) is a more conserved quantity than $O_3$ and hence a better metric for comparison (Kley et al., 1994; Kleinmann et al., 2002; Caputi et al., 2019; He et al., 2022). During daytime, $NO_2$ formed in the titration reaction is rapidly photolyzed to regenerate $O_3$ so that the net chemical loss of $O_3$ is relatively small. At night, however, the reaction between $O_3$ and NO leads to lower levels of $O_3$ in the nocturnal boundary layer (NBL) and in the residual layer (RL). Because the emission of NO takes place mainly in the NBL, much more $O_3$ is removed by the titration reaction in the NBL than in the RL (Wang et al., 2018; Caputi et al., 2019; He et al., 2022). In addition, $O_3$ in the NBL is subjected to dry deposition. Therefore, the $O_3$ level before sunset largely remains in the RL (Caputi et al., 2019; He et al., 2022) and is usually much higher in the RL than in the NBL under normal conditions.

Following the method of He et al. (2022), we make comparison of afternoon $O_3$ averages on 31 July with the respective $O_x$ averages in the NBL during the NOE between 31 July and 1 August, 2021. To facilitate the comparison, we treat the average surface $O_3$ during 14:00-17:00 LT of 31 July as afternoon average of $O_3$ in the convective boundary layer, denoted as $[O_3]_{aft}$. Let us now focus on three nighttime atmospheric conditions, (I) undisturbed, (II) disturbed with NOE but no STT impact, and (III) disturbed with NOE and significant STT impact.

Under undisturbed condition (I), the nighttime average $O_3$ concentration in the RL ($[O_3]_{RL}$) should be close to (or only slightly lower than) $[O_3]_{aft}$ (Caputi et al., 2019; He et al., 2022), while the average $O_3$ concentration in the NBL ($[O_3]_{NBL}$) should be much lower than $[O_3]_{aft}$ due to the impacts of NO titration ($\Delta[O_3]_{titr}$) and dry deposition ($\Delta[O_3]_{dep}$), and the average $O_x$ concentration in the NBL ($[O_x]_{NBL}$) should also be lower than $[O_3]_{aft}$ due to dry deposition. The following relationships should be tenable:

$$[O_3]_{RL} \leq [O_3]_{aft} \tag{S1}$$

$$[O_3]_{NBL} = [O_3]_{aft} - \Delta[O_3]_{titr} - \Delta[O_3]_{dep} \tag{S2}$$

$$[O_x]_{NBL} = [O_3]_{aft} - \Delta[O_3]_{dep} \tag{S3}$$

Under disturbed condition with NOE but no STT impact (II), a downward transport of $O_3$ from the RL to NBL should be considered. Assuming that the downward transport causes a reduction of $[O_3]_{RL}$ by $\Delta[O_3]_{D1}$ and an increase of $[O_3]_{NBL}$ by $\Delta[O_3]_{D2}$, then

$$[O_3]_{RL} \leq [O_3]_{aft} - \Delta[O_3]_{D1} \tag{S4}$$

$$[O_3]_{NBL} = [O_3]_{aft} - \Delta[O_3]_{titr} - \Delta[O_3]_{dep} + \Delta[O_3]_{D2} \tag{S5}$$

$$[O_x]_{NBL} = [O_3]_{aft} - \Delta[O_3]_{dep} + \Delta[O_3]_{D2} \tag{S6}$$

Under disturbed condition with NOE and STT impact (III), net contributions of $O_3$ from the STT should be considered to the RL and the NBL. Assuming that the STT contribution increases $[O_3]_{RL}$ and $[O_3]_{NBL}$ by $\Delta[O_3]_{STT1}$ and $\Delta[O_3]_{STT2}$,

respectively, then

$$[O_3]_{RL} \leq [O_3]_{aft} + \Delta[O_3]_{STT1} \quad\quad\quad\quad\quad\quad\quad\quad\quad\quad\quad\quad\quad (S7)$$

$$[O_3]_{NBL} = [O_3]_{aft} - \Delta[O_3]_{titr} - \Delta[O_3]_{dep} \pm \Delta[O_3]_{STT2} \quad\quad\quad\quad\quad\quad (S8)$$

$$[O_x]_{NBL} = [O_3]_{aft} - \Delta[O_3]_{dep} + \Delta[O_3]_{STT2} \quad\quad\quad\quad\quad\quad\quad\quad\quad (S9)$$

Equation (S3) indicates that $[O_x]_{NBL}$ should be significantly lower than $[O_3]_{aft}$ under undisturbed conditions. Although equation (S6) shows that $[O_x]_{NBL}$ could be higher than $[O_3]_{aft}$ (i.e., if $\Delta[O_3]_{dep} < \Delta[O_3]_{D2}$), it cannot really occur because $[O_3]_{RL} < [O_3]_{aft}$ (see equation (S4)) and $O_3$ cannot be transported from a lower concentration position to the higher one. Therefore, $[O_x]_{NBL}$ should not be significantly higher than $[O_3]_{aft}$ under disturbed conditions with NOE but no STT impact. Dry deposition is only a small sink for nighttime surface $O_3$ in northern China (Tang et al., 2017), while a STT impact could substantially enhanced the level of surface $O_3$ if it reaches the surface layer. Hence, it is very likely according to equation (S9) that $[O_x]_{NBL}$ is significantly higher than $[O_3]_{aft}$ under disturbed condition with a STT impact. In summary, $[O_x]_{NBL}$ should be significantly higher than $[O_3]_{aft}$ if the NBL is really impacted by stratospheric $O_3$, otherwise the STT impact is negligible even though a NOE event is observed.

[revised manuscript text omitted]